# HARPOON: GENERALISED MANIFOLD GUIDANCE FOR CONDITIONAL TABULAR DIFFUSION

**Aditya Shankar[1]  Yuandou Wang[1]  Rihan Hai[1]  Lydia Chen[1,2]**
[1]Department of Computer Science, Delft University of Technology
[2]Department of Computer Science, Université de Neuchâtel
`{a.shankar,y.wang-45,r.hai}@tudelft.nl`
`lydiaychen@ieee.org`

## ABSTRACT

Generating tabular data under conditions is critical to applications requiring precise control over the generative process. Existing methods rely on training-time strategies that do not generalise to unseen constraints during inference, and struggle to handle conditional tasks beyond tabular imputation. While manifold theory offers a principled way to guide generation, current formulations are tied to specific inference-time objectives and are limited to continuous domains. We extend manifold theory to tabular data and expand its scope to handle diverse inference-time objectives. On this foundation, we introduce HARPOON, a tabular diffusion method that guides unconstrained samples along the manifold geometry to satisfy diverse tabular conditions at inference. We validate our theoretical contributions empirically on tasks such as imputation and enforcing inequality constraints, demonstrating HARPOON'S strong performance across diverse datasets and the practical benefits of manifold-aware guidance for tabular data. Code URL: `https://github.com/adis98/Harpoon`

## 1 INTRODUCTION

Generating tabular data subject to *conditions* (constraints) is critical to many tasks, such as *imputing* missing values given partial observations (Zhang et al., 2025) or simulating hypothetical "*what-if*" scenarios for decision-making (Narasimhan et al., 2024). Such tasks require strong generative models, among which *diffusion* models (Ho et al., 2020; Song et al., 2021) have emerged as state-of-the-art (SOTA) across many modalities (Dhariwal & Nichol, 2021; Kotelnikov et al., 2023; Kollovieh et al., 2023). However, *conditional* tabular diffusion models (Zheng & Charoenphakdee, 2022; Ouyang et al., 2023; Villaizán-Vallelado et al., 2025; Zhang et al., 2025) often rely on training-time strategies that cannot generalise to unseen conditions at inference, or cannot efficiently handle diverse objectives, such as inequality constraints on features (e.g. `Age >= 10`) (Stoian et al., 2024).

Recent works in image diffusion offer a geometric view of the diffusion process (Chung et al., 2022; He et al., 2024), treating data as lying on a low-dimensional *manifold* (Gorban & Tyukin, 2018; Bengio et al., 2013; Pope et al., 2021). They show that naively enforcing conditions can push generated samples off the manifold, producing unrealistic results. These works propose constraining samples to stay close to the manifold's surface until converging to the desired conditions. However, these theories are limited to continuous domains and assume *flat* geometries, which may not hold for mixed-type tabular data. They also only analyse specific inference-time losses (e.g., squared-error) without generalising to diverse objectives, such as inequality constraints.

We extend manifold theory to tabular diffusion models operating in the data space. We first prove that diffusion models implicitly learn an *orthogonal* mapping onto the underlying manifold when trained using the standard squared-error loss, even without strong assumptions on the curvature. We then prove that the gradient of *any* differentiable inference-time loss lies in the *tangent space* of the manifold. With this powerful result, we introduce HARPOON, a conditional tabular diffusion method that requires training *once* and adapts to diverse conditions at *inference*. HARPOON interleaves tangential gradient corrections with unconstrained denoising steps, gradually guiding samples along the

manifold's surface towards regions satisfying diverse inference-time objectives, such as imputation or inequality constraints on features. We summarise our contributions as follows:

**Theory.** We provide the *first theoretical results* linking manifolds to tabular diffusion, including curved geometries. We prove that diffusion models act as orthogonal projectors, guaranteeing the tangent-space behaviour of gradients from *any* differentiable inference-time objective.

**Algorithm.** Building on our theoretical results, we design a manifold-aware conditional tabular diffusion method, HARPOON, that guides unconstrained samples towards satisfying diverse inference-time tabular generative constraints.

**Empirical validation.** We empirically validate the geometric alignment of the inference-time gradients with the underlying manifold, demonstrating HARPOON'S strong performance across diverse datasets and conditional tasks, including imputation and inequality conditions.

## 2 BACKGROUND AND RELATED WORK

We review diffusion models for tabular data and their conditional extensions. We then turn towards manifold-based perspectives from image diffusion to motivate our contributions. The key notations used from this section onwards are summarised in Appendix A.

**Tabular Diffusion.** Denoising Diffusion Probabilistic Models (DDPMs) (Ho et al., 2020) define a generative procedure by reversing a Markovian noising process, $p(x_1, \ldots, x_T|x_0) = \prod_{t=1}^{T} p(x_t|x_{t-1})^1$, which progressively corrupts a clean sample $x_0$ over $T$ steps. For tabular data containing a mix of discrete and continuous features, $x_0$ is represented as $\left[x_0^{\text{cont}}, x_0^{(1)}, \ldots, x_0^{(k)}\right]$, with continuous features $x_0^{\text{cont}}$ and categorical vectors $x_0^{(i)}$ that can be represented using one-hot, integer, or binary encoding schemes. The standard approach for the *forward* (noising) process across data modalities is to add continuous Gaussian noise to samples, as follows:

$$x_t = \sqrt{\bar{\alpha}_t}\, x_0 + \sqrt{1 - \bar{\alpha}_t}\, \epsilon, \quad \epsilon \sim \mathcal{N}(0, I), \tag{1}$$

where $\bar{\alpha}_t = \Pi_{s=1}^{t} \alpha_s$ is a cumulative noise schedule, with $\bar{\alpha}_t \to 0$ as $t \to T$ (Ho et al., 2020). The resulting noisy distribution is $p(x_t) := \int p(x_t|x_0)p(x_0)dx_0$ with $p(x_t|x_0) \sim \mathcal{N}(\sqrt{\bar{\alpha}_t}x_0, (1-\bar{\alpha}_t)I)$.

The *backward* process iteratively reconstructs $x_0$ from $x_T$ using a neural network (a.k.a *denoiser*) with parameters $\theta$ that predicts $p_\theta(x_{t-1}|x_t)$ at each step by estimating the added noise $\epsilon$. The standard training loss is the Mean-Squared-Error (MSE) (Ho et al., 2020) using estimates $\epsilon_\theta(x_t, t)$:

$$\mathcal{L}_{\text{train}} = \mathbb{E}_{x_0 \sim p(x_0),\, t \sim U[1,T],\, \epsilon \sim \mathcal{N}(0,I)} \left[\left\|\epsilon - \epsilon_\theta(x_t, t)\right\|_2^2\right], \tag{2}$$

For tabular data, the general standard is to model continuous and discrete variables using separate diffusion processes, such as *masked* (Shi et al., 2025; Austin et al., 2021) or *multinomial* (Kotelnikov et al., 2023; Hoogeboom et al., 2021) diffusion for categorical features. However, these approaches introduce additional training complexity, requiring separate loss functions, such as MSE and cross-entropy for continuous and discrete features, respectively. Another alternative is to stick with a single continuous diffusion process and squared-error objective, but instead apply it on smooth latent space embeddings of samples using autoencoders (Zhang et al., 2024; Shankar et al., 2024).

**Conditional Tabular Diffusion.** Conditional generation introduces a variable $c$ to guide the generation process. We use $c$ to broadly denote any user-specified constraint, ranging from binary masks that indicate partially observed entries in imputation tasks (Xu et al., 2019; Alcaraz & Strodthoff, 2022) to inequality constraints on features (Stoian et al., 2024). A key challenge is that these conditions often correspond to *diverse* and *non-linear* objectives (Donti et al., 2021). As a result, enforcing them at inference scales poorly, since existing approaches typically require separate retraining or fine-tuning of the generative model for each new condition.

Conditional tabular diffusion methods typically fall into two categories: *training-time* and *inference-time* solutions. Training-time methods learn the conditional distribution $p_\theta(x_{t-1}|x_t, c)$ during training, using $c$ as model inputs (Zheng & Charoenphakdee, 2022; Villaizán-Vallelado et al., 2025; Ouyang et al., 2023). However, this limits generalisation to unseen conditions at inference. *Classifier guidance* (Dhariwal & Nichol, 2021), trains an auxiliary classifier to predict the gradient

---

[1]We use $p$ for the forward process; $p_\theta$ denotes the learned denoising process parameterized by $\theta$.

$\nabla_{x_t} \log p(c|x_t)$, to steer sampling (Liu et al., 2024). While this is effective when $c$ denotes discrete labels, it is not applicable to real-valued constraints. Moreover, the pre-trained classifiers cannot guide samples towards conditions unseen during training. *Classifier-free guidance* (Ho & Salimans, 2021) avoids the need for separate classifiers by repurposing the diffusion backbone. It jointly trains and interpolates between a conditional and unconditional model to control the degree of guidance. However, it is still limited by the conditional branch, which requires training on constraints known in advance, affecting inference-time generalisation.

In contrast, inference-time methods for imputation (Zhang et al., 2025) guide generation by forward noising observed values and jointly denoising observed and missing values, similar to image inpainting (Lugmayr et al., 2022). While effective for imputation, they require partially observed ground truths and cannot handle inequality constraints. Rejection sampling (Casella et al., 2004) is a general fallback, but is impractical due to high violation rates.

Finally, we note that conditional generative methods typically operate in the *data* space, where constraints are specified directly on features. Conditioning latent methods is challenging because it requires translating data space constraints into their latent counterparts, which is only possible when there is a deterministic mapping between the two spaces (Zhang et al., 2024).

**The Geometry of (Image) Diffusion.** Recent works in image diffusion interpret the diffusion process as a sequence of *manifold transitions* (Chung et al., 2022; He et al., 2024). A clean sample $x_0$ is assumed to lie on a *low-dimensional* manifold $\mathcal{M}_0 \subset \mathbb{R}^d$. The forward process gradually perturbs $\mathcal{M}_0$ into a family of surrounding *shells* $\{\mathcal{M}_t\}_{t=1}^T$, where $\mathcal{M}_T$ approximates pure Gaussian noise (see Figure 1). Noisy samples concentrate near these shells, which the model learns to denoise. Enforcing constraints by pushing samples orthogonally toward $\mathcal{M}_0$ risks "skipping" shells, yielding off-distribution inputs where model estimates become unreliable (Chung et al., 2022). Crucially, Chung et al. (2022) show that the gradients of inference-time squared-error objectives lie in the *tangent* space of $\mathcal{M}_0$, and leverage these tangential corrections to help keep samples within the current shell while steering them *along* the manifold geometry to enable condition-aware denoising.

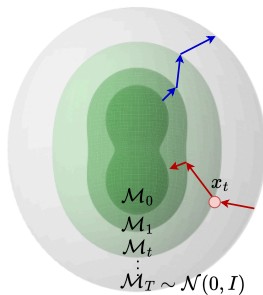

Figure 1: Geometry of forward (↑) and backward (↑) diffusion.

Extending these insights to tabular diffusion is not trivial. First, existing theories assume continuous features, while tabular models have both continuous and discrete features (Hoogeboom et al., 2021; Kotelnikov et al., 2023), thereby violating these assumptions. Second, current theoretical proofs implicitly assume *flat* manifold geometries (Chung et al., 2022; He et al., 2024), which need not hold for mixed-type tabular data. Third, the tangent-space guarantees only hold for *squared-error* losses (Theorem 1, Chung et al. (2022)). In contrast, tabular conditions, such as inequalities or imputation masks, need *diverse*, often *non-linear* differentiable losses (see eqs. 7, 9, and 10, Appendix D), which is beyond the current theoretical scope.

## 3 MANIFOLD GUIDANCE FOR TABULAR DIFFUSION

We first derive the theoretical foundations for HARPOON and then detail its algorithm for conditional tabular generation. We define the necessary notations and assumptions, leading up to our main theoretical contribution: a manifold guidance technique for *any differentiable inference-time loss*.

### 3.1 PRELIMINARIES AND ASSUMPTIONS

We represent a tabular sample $x_0 \in \mathbb{R}^d$ as a concatenation of continuous and one-hot encoded categorical features, same as Sec. 2. We assume that the support of the data distribution $p(x_0)$ lies on a continuous smooth manifold $\mathcal{M}_0 \subset \mathbb{R}^d$ (see Remark 1). Here, $d$ denotes the ambient space dimension, and $n = \dim(\mathcal{M}_0)$ is the intrinsic dimension of the data manifold, with $n \ll d$. We formalise this with standard geometric assumptions (Chung et al., 2022).

**Assumption 1** (Smooth Embedding). *The support of the clean data distribution $p(x_0)$ lies on a smooth, connected, $n$-dimensional manifold $\mathcal{M}_0$ embedded in the ambient space $\mathbb{R}^d$, where $n \ll d$. Formally, there exists a smooth embedding $\varphi : \mathbb{R}^n \to \mathbb{R}^d$ such that $\mathcal{M}_0 = \varphi(\mathbb{R}^n)$.*

**Assumption 2** (Local Linearity). *For any point $x_0 \in \mathcal{M}_0$, the manifold is locally well-approximated by its tangent space $T_{x_0}\mathcal{M}_0$ at $x_0$. That is, there exists a radius $r > 0$ such that within the hypersphere $B_r(x_0) = \{x \in \mathbb{R}^d : \|x - x_0\|_2 < r\}$, the manifold coincides with its tangent plane:*

$$\mathcal{M}_0 \cap B_r(x_0) = T_{x_0}\mathcal{M}_0 \cap B_r(x_0).$$

*Intuitively, sufficiently zooming in around $x_0$ makes the manifold appear locally flat.*

*Remark* 1. [**Tabular representation**] A sample $x_0 \in \mathbb{R}^d$ consists of continuous and categorical variables in the tabular setting. To extend Assumptions 1 and 2 to tabular data, we assume the support of the encoded points resides in a continuous space. For example, a discrete feature with $K$ classes can be encoded as a one-hot vector, or as a "soft" probability distribution over the $K$ classes. This relaxes categories into points within the $(K$-$1)$-simplex, giving a continuous embedding. While such a continuous approximation may be suboptimal, we later show that our theoretical derivations provide a way to overcome this limitation for tabular diffusion.

Building on these assumptions, we connect the forward process to the geometry of noisy manifolds. From eq. 1, , each $x_t$ that is the outcome of the forward process can be viewed as lying near a scaled copy of the clean data manifold, perturbed by Gaussian noise (see Figure 1). We formalise this intuition in Proposition 1, formally proved in He et al. (2024).

**Proposition 1** (Shell Structure). *During the forward diffusion process, noisy samples $x_t$ at diffusion step $t$ probabilistically concentrate on a $(d-1)$-dimensional manifold $\mathcal{M}_t$ forming a shell around a scaled copy of the $n$-dimensional data manifold $\mathcal{M}_0$:*

$$\mathcal{M}_t := \{x \in \mathbb{R}^d \mid d(x, \sqrt{\bar{\alpha}_t}\mathcal{M}_0) = \sqrt{(1 - \bar{\alpha}_t)(d - n)}\}.$$

Here, $d(x, \sqrt{\bar{\alpha}_t}\mathcal{M}_0) := \inf_{x' \in \mathcal{M}_0} \|x - \sqrt{\bar{\alpha}_t}x'\|_2$ denotes the Euclidean distance from $x$ to the closest point on the scaled manifold $\sqrt{\bar{\alpha}_t}\mathcal{M}_0$. Note that $\bar{\alpha}_t \to 1$ as $t \to 0$. As $t$ increases, the shells drift further away from $\mathcal{M}_0$, giving the structure shown in Figure 1.

Under the standard training objective in eq. 2, the model learns to denoise points concentrated on the noisy manifolds $\mathcal{M}_t$, iteratively reconstructing clean data $x_0$ from Gaussian noise.

## 3.2 GENERALISED MANIFOLD THEORY

Based on the Gaussian noising process and denoiser training objective introduced in Sec. 3.1, we now develop a generalised manifold theory for conditional tabular diffusion. We first define the denoiser mapping $Q_t$, which predicts dirty estimates from noisy samples $x_t$ by reversing equation 1.

**Definition 1** ("Dirty" Estimate via Diffusion Model). *Let $\epsilon_\theta(x_t, t)$ denote the diffusion model's noise estimate. We define the dirty estimate of a noisy sample $x_t$ by reversing eq. 1 (forward process).*

$$\hat{x}_0 := Q_t(x_t) := \frac{1}{\sqrt{\bar{\alpha}_t}}\big(x_t - \sqrt{1 - \bar{\alpha}_t}\,\epsilon_\theta(x_t, t)\big), \tag{3}$$

*where $Q_t : \mathbb{R}^d \to \mathbb{R}^d$ maps a noisy input to a point $\hat{x}_0 \in \mathcal{M}_0$. Intuitively, $Q_t(x_t)$ lies near the clean data manifold $\mathcal{M}_0$, since it is a reversal of the forward process (eq. 1) that noises a clean sample.*

We show that for a model trained under a Gaussian noising process using eq. 2, $Q_t$ naturally aligns with the *orthogonal projection* of $x_t$ onto $\mathcal{M}_0$, as formalised in the following theorem.

**Theorem 3.1** (Limiting behaviour of dirty estimates). *Let $\epsilon_\theta$ be a denoiser trained to predict Gaussian noise under the framework of eqs. 1 and 2, for a clean sample $x_0$. Then, for the mapping $Q_t$ defined in Definition 1, we have $\lim_{\bar{\alpha}_t \to 1} Q_t(x_t) = \pi(x_t)$, where $\pi(x_t)$ denotes the orthogonal projection of $x_t$ onto the data manifold $\mathcal{M}_0$.*

The formal proof is in Appendix B. Theorem 3.1 establishes a general and practically relevant guarantee that the dirty estimates $Q_t(x_t)$ map to the orthogonal projection $\pi(x_t)$ on the data manifold as $\bar{\alpha}_t \to 1$. Figure 2a intuitively explains this behaviour, where the "*spotlight*" for the dirty estimate's density distribution sharpens around the orthogonal projection as $x_t$ approaches $\mathcal{M}_0$.

We note that Chung et al. (2022) present a related claim (their Proposition 2), which does not require the limiting case $\bar{\alpha}_t \to 1$. This is because their proof *implicitly* assumes the manifold is *globally*

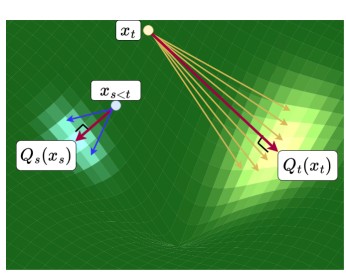
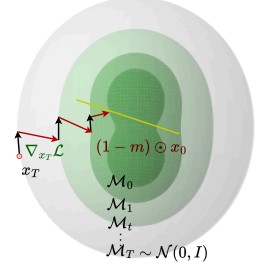


(a) Spotlight of $x_t$'s projections     (b) Imputation constraints     (c) Inequality conditions

Figure 2: (a) Shows the orthogonal behaviour of dirty estimates projected from $x_t$ to $\hat{x}_0$. Figs. (a) and (b) show HARPOON'S guidance mechanism, interleaving unconditional denoising ($\rightarrow$) and tangential updates ($\uparrow$) for (b) imputation constraints of the form $(1 - m) \odot x_0$, and (c) inequality constraints. Yellow regions indicate the areas on the manifold $\mathcal{M}_0$ matching the constraint.

*flat*, allowing them to invoke symmetry arguments to explain the orthogonal behaviour (detailed in Appendix B). We *generalise* their result, proving that the orthogonal behaviour arises even without such restrictive assumptions, but in the limit as $\bar{\alpha}_t \to 1$. We use Theorem 3.1 to rigorously justify the assumptions used in the following theorem.

**Theorem 3.2** (Manifold-constrained inference-time gradients). *Let $c$ denote arbitrary conditioning information and assume $Q_t$ from Definition 1 acts as an orthogonal projection onto the data manifold $\mathcal{M}_0$ at $\hat{x}_0 = Q_t(x_t)$. Then, for any differentiable inference-time loss function $\mathcal{L}_{\inf} : \mathbb{R}^d \times \mathcal{C} \to \mathbb{R}$, defined by $\mathcal{L}_{\inf}(\hat{x}_0; c)$, the gradient lies in the tangent space of $\mathcal{M}_0$ at $\hat{x}_0$, i.e., $\nabla_{x_t} \mathcal{L}_{\inf}(\hat{x}_0, c) \in T_{\hat{x}_0} \mathcal{M}_0$.*

We provide the proof in Appendix B. Theorem 3.2 establishes a powerful result that the gradients of *any* differentiable objective at inference ($\mathcal{L}_{\inf}$) are aligned with the tangent space of the data manifold. This generalises prior work, which proved the tangential behaviour only for squared-error inference-time losses of the form $||W(x_0 - H(\hat{x}_0))||_2^2$ under restrictive symmetry and manifold-linearity assumptions (Theorem 1, Chung et al. (2022)). Our formulation is strictly more general, covering curved geometries. It applies to *arbitrary* differentiable losses, so supports a rich spectrum of inference conditions $c$, ranging from partially observed values (e.g. imputation masks) to prompts, metadata, or inequality constraints. Importantly, the inference-time losses ($\mathcal{L}_{\inf}$) can be different from the training loss ($\mathcal{L}_{\text{train}}$, eq. 2).

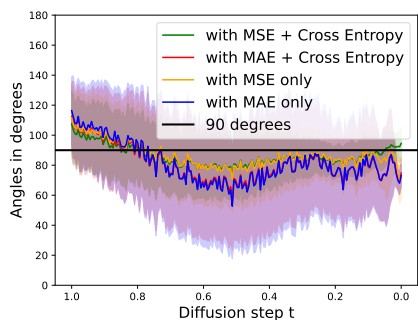

Figure 3: Avg. angle between gradients (100 samples) and dirty estimates $\hat{x}_0$ for various loss functions on Adult.

**Tabular adaptation.** Theorem 3.2 is critical for tabular data. Even if training relies on a continuous Gaussian noising framework to avoid separating discrete and continuous diffusion, inference can incorporate losses tailored to discrete variables, such as cross-entropy, inequality constraints, or even $L_1$-based penalties with sparsity-inducing properties Bach et al. (2012) suited to tabular domains.

**Empirical orthogonality.** Strictly speaking, the tangential behaviour of gradients requires the dirty estimate $Q_t(x_t)$ to be orthogonal, which is guaranteed only in the limiting case $\bar{\alpha}_t \to 1$, i.e., as $t \to 0$. However, in practice, we observe that the gradients remain close to $90°$ with $Q_t(x_t)$ even at larger time steps, suggesting that the orthogonality property extends well beyond the limit (see Figure 3). Additionally, Figure 3 shows a consistent behaviour across varying inference-time losses under the same training objective (MSE), giving an empirical verification of Theorem 3.2.

Theorems 3.1 and 3.2, together with the observations in Figure 3, directly motivate HARPOON'S design, which functions like a high-precision guidance system. The denoiser estimate $Q_t(x_t)$ acts like

a *compass*, pointing toward the orthogonal projection onto $\mathcal{M}_0$. Meanwhile, the manifold-aligned gradients act as *anchors*, steering updates *tangentially* along the manifold surface, keeping samples within the manifold shells for reliable model estimates. The idea is analogous to a ship navigating against a current through steady movements along the manifold without drifting off course.

Leveraging this intuition, HARPOON interleaves *unconditional* denoising steps with tangential gradient corrections, *guiding* samples towards the specified conditions. Since consecutive manifolds are nearly parallel (Proposition 1), we can (unconditionally) denoise one step followed by a tangential correction using gradients from the previous step, gradually converging towards the target region (see Figs. 2b,2c). This guidance system naturally adapts to a variety of tabular conditions. For inequality constraints (Figure 2c), the target areas span a region on the manifold's surface, and Harpoon guides samples to *"land"* within this region. For imputation-style tasks (Figure 2b), the constraints instead anchor samples to the partial observations on the manifold of the form $(1-m) \odot x_0$ (see Appendix D), lying on a constraint line for a given missing value binary mask $m$.

At a high level, this interleaved sampling is structurally similar to prior training-free guidance methods (Chung et al., 2023; 2022; He et al., 2024; Bansal et al., 2023; Song et al., 2023; Yu et al., 2023). The distinction is in the class of theoretically justified objectives. Existing work either justifies guidance only under restrictive flat-manifold assumptions (Chung et al., 2022; 2023; He et al., 2024), or by applying inference-time losses without analysis of geometric compatibility with the underlying manifold (Song et al., 2023; Bansal et al., 2023; Yu et al., 2023). We remove this dichotomy by generalising the tangent-space behaviour of guidance even to arbitrary inference-time objectives.

### 3.3 HARPOON ALGORITHM

We concretely detail the guidance process in Algorithm 1. At each step $t$, we first compute the dirty estimate using equation 3. We then pre-compute the tangential gradient corrections by applying the the inference-time objective ($\mathcal{L}_{\inf}$) on the dirty estimates, i.e., $\nabla_{x_t} \mathcal{L}_{\inf}(\hat{x}_0, c)$ (line 6). Then, the sample is *unconditionally* denoised one step using the standard reverse process (Ho et al., 2020) (line 8). As the denoising step is unconstrained, we apply the pre-computed tangential correction to guide the sample towards the objective using a guidance strength $\eta$ (line 9).

Importantly, the inference-time loss $\mathcal{L}_{\inf}(\hat{x}_0, c)$ can be different from the training objective, including imputation

---

**Algorithm 1** Sampling with HARPOON

1: Initialize $x_T \sim \mathcal{N}(0, I)$
2: **for** $t = T, \ldots, 1$ **do**
3:     Sample $z \sim \mathcal{N}(0, I)$ if $t > 1$, else $z = 0$
4:     Estimate noise $\epsilon_\theta(x_t, t)$
5:     Obtain dirty estimate $\hat{x}_0$ (eq. 3)
6:     Get tangential gradient $g = \nabla_{x_t} \mathcal{L}_{\inf}(\hat{x}_0, c)$
7:     Set $\sigma_t = \frac{(1-\alpha_t).(1-\bar{\alpha}_{t-1})}{(1-\bar{\alpha}_t)}$
8:     Denoise one step:
$$x'_{t-1} = \frac{1}{\sqrt{\alpha_t}} \left( x_t - \frac{1-\alpha_t}{\sqrt{1-\bar{\alpha}_t}} \epsilon_\theta(x_t, t) \right) + \sigma_t z$$
9:     Apply tangential update:
$$x_{t-1} = x'_{t-1} - \eta \cdot g$$
10: **return** $x_0$

---

or ReLU-based inequality losses (Donti et al., 2021) (see Appendix D), handling a rich variety of generative tabular conditions. Moreover, since tabular data is represented using sparse one-hot encoded discrete features, we use the *sparsity-inducing* Mean-Absolute-Error (MAE) loss for improved convergence at inference time by default (Bach et al., 2012).

## 4 EXPERIMENTS

We evaluate the practical benefits of HARPOON's manifold-guided updates for conditional generation across two representative tasks: (i) *imputation* of missing values from partially observed samples, and (ii) *inequality constraints*, which enforce conditions without relying on ground-truth targets. We further analyse the effect of tabular-specific losses at inference time, study different guidance schedules, and measure runtime efficiency. Additional experiments, covering alternative mask patterns, encoding schemes, and evaluation metrics, are reported in Appendix I.

### 4.1 EXPERIMENTAL SETTINGS

**Datasets.** We evaluate on eight widely used benchmarks, mostly from the UCI repository (Dua et al., 2017). Five with only continuous features (Gesture, Magic, California, Letter, and Bean), and

Table 1: Imputation MSE on continuous features for MAR mask. Standard deviation in subscript.

| Ratio | Method | Adult | Bean | California | Default | Gesture | Letter | Magic | Shoppers |
|---|---|---|---|---|---|---|---|---|---|
| 0.25 | GAIN | $1.86_{0.04}$ | $1.41_{0.03}$ | $15.06_{4.32}$ | $6.86_{1.61}$ | $1.22_{0.08}$ | $1.05_{0.04}$ | $1.27_{0.24}$ | $1.67_{0.26}$ |
| | Miracle | $2.17_{0.34}$ | $0.31_{0.12}$ | $0.50_{0.11}$ | $1.51_{0.38}$ | $0.94_{0.10}$ | $1.32_{0.11}$ | $0.88_{0.14}$ | $2.21_{0.76}$ |
| | GReaT | $3.70_{3.26}$ | $1.82_{0.05}$ | $1.55_{0.05}$ | $11.39_{15.12}$ | $>10^5$ | $1.67_{0.04}$ | $1.64_{0.15}$ | $1.51_{0.10}$ |
| | Remasker | $\mathbf{0.92}_{0.04}$ | $0.23_{0.06}$ | $0.89_{0.03}$ | $\underline{0.56}_{0.05}$ | $0.70_{0.06}$ | $0.57_{0.03}$ | $\underline{0.81}_{0.11}$ | $\mathbf{0.72}_{0.12}$ |
| | DiffPuter | $1.04_{0.09}$ | $\underline{0.18}_{0.03}$ | $0.47_{0.04}$ | $0.62_{0.08}$ | $\mathbf{0.40}_{0.07}$ | $0.42_{0.04}$ | $0.77_{0.16}$ | $0.76_{0.10}$ |
| | HARPOON | $\underline{0.99}_{0.08}$ | $\mathbf{0.15}_{0.06}$ | $\mathbf{0.42}_{0.04}$ | $\mathbf{0.51}_{0.06}$ | $\underline{0.51}_{0.09}$ | $\mathbf{0.52}_{0.04}$ | $0.97_{0.29}$ | $\underline{0.73}_{0.06}$ |
| 0.50 | GAIN | $3.37_{0.21}$ | $1.94_{0.12}$ | $18.69_{5.33}$ | $23.17_{7.05}$ | $2.97_{0.32}$ | $1.11_{0.03}$ | $1.36_{0.45}$ | $3.86_{1.20}$ |
| | Miracle | $2.08_{0.40}$ | $0.49_{0.06}$ | $0.92_{0.07}$ | $1.03_{0.19}$ | $1.04_{0.09}$ | $0.83_{0.13}$ | $1.07_{0.14}$ | $1.36_{0.23}$ |
| | GReaT | $1.61_{0.21}$ | $4.94_{4.65}$ | $1.41_{0.19}$ | $1.84_{1.12}$ | $>10^5$ | $1.39_{0.04}$ | $1.24_{0.11}$ | $>10^5$ |
| | Remasker | $\mathbf{0.94}_{0.03}$ | $0.42_{0.05}$ | $0.90_{0.05}$ | $\underline{0.59}_{0.05}$ | $0.97_{0.08}$ | $\underline{0.78}_{0.03}$ | $\mathbf{0.77}_{0.08}$ | $\underline{0.82}_{0.10}$ |
| | DiffPuter | $1.18_{0.08}$ | $\underline{0.28}_{0.06}$ | $\underline{0.79}_{0.12}$ | $0.64_{0.07}$ | $\mathbf{0.48}_{0.06}$ | $0.75_{0.04}$ | $\underline{0.80}_{0.08}$ | $0.88_{0.13}$ |
| | HARPOON | $\underline{1.08}_{0.06}$ | $\mathbf{0.19}_{0.05}$ | $\mathbf{0.72}_{0.07}$ | $\mathbf{0.53}_{0.06}$ | $\underline{0.63}_{0.09}$ | $0.86_{0.05}$ | $0.86_{0.16}$ | $\mathbf{0.81}_{0.06}$ |
| 0.75 | GAIN | $7.44_{0.21}$ | $3.10_{0.07}$ | $22.74_{1.33}$ | $62.23_{10.77}$ | $9.70_{1.17}$ | $1.29_{0.03}$ | $2.04_{0.59}$ | $14.85_{1.52}$ |
| | Miracle | $1.27_{0.04}$ | $0.75_{0.06}$ | $\mathbf{0.98}_{0.02}$ | $>10^9$ | $1.23_{0.05}$ | $\mathbf{1.00}_{0.02}$ | $\underline{1.02}_{0.03}$ | $\underline{1.11}_{0.00}$ |
| | GReaT | $1.70_{0.48}$ | $4.55_{2.46}$ | $1.43_{0.06}$ | $2.10_{0.58}$ | $>10^5$ | $1.44_{0.04}$ | $1.19_{0.04}$ | $1.28_{0.02}$ |
| | Remasker | $\mathbf{1.03}_{0.01}$ | $1.14_{0.07}$ | $1.13_{0.06}$ | $\underline{0.89}_{0.05}$ | $3.98_{0.63}$ | $\underline{1.15}_{0.02}$ | $1.30_{0.10}$ | $1.17_{0.03}$ |
| | DiffPuter | $1.49_{0.09}$ | $\underline{0.61}_{0.04}$ | $1.10_{0.08}$ | $\underline{0.89}_{0.10}$ | $\mathbf{0.71}_{0.03}$ | $1.22_{0.04}$ | $1.08_{0.09}$ | $1.26_{0.08}$ |
| | HARPOON | $\underline{1.26}_{0.07}$ | $\mathbf{0.37}_{0.03}$ | $\underline{1.02}_{0.07}$ | $\mathbf{0.72}_{0.06}$ | $\underline{0.84}_{0.05}$ | $1.30_{0.05}$ | $\mathbf{1.00}_{0.08}$ | $\mathbf{1.10}_{0.05}$ |

three with both discrete and categorical features (Adult, Default, and Shoppers). The preprocessing steps are detailed in Appendix C. Each dataset is randomly split into 70% training and 30% test sets.

**Baselines.** We compare against SOTA tabular methods spanning causal graphs, deep learning, generative, and large language model based (LLM) approaches. The baselines include: (i) MIRA-CLE (Kyono et al., 2021), a causal graph-based imputation method, (ii) DiffPuter (Zhang et al., 2025), a SOTA conditional tabular diffusion model, (iii) GReaT (Borisov et al., 2023), an LLM-based method for tabular generation and imputation, (iv) Remasker (Du et al., 2024), a masked autoencoder imputation method, and (v) GAIN (Yoon et al., 2018), a conditional tabular GAN model. For fairness, all diffusion-based approaches, including ours, share the same backbone architecture. Details on the baseline implementations and hyperparameters are provided in Appendix G.

**Evaluation Protocol.** Since our goal is to enable conditioning at inference, all models are pretrained *once* and then evaluated strictly at inference time across different tasks, including imputation and inequality conditions. For the imputation tasks, we evaluate under missingness ratios of 0.25, 0.5, and 0.75. Continuous features are evaluated using the Mean-Squared-Error (MSE) between imputed values and the ground truths, while categorical features are assessed using the match accuracy. We consider three missingness patterns following prior work (Muzellec et al., 2020): *Missing-At-Random* (MAR) (main text), and *Missing-Completely-At-Random* (MCAR) and Missing-Not-At-Random (MNAR) (Appendix I.1). Dataset preprocessing and mask generation details are provided in Appendix E. For the general inequality conditions, we consider four scenarios: (i) *range* constraints on a continuous feature (e.g. $12 > Age >= 10$); (ii) *categorical* constraints, where a discrete feature is fixed to a target value (e.g. *Gender* = 'Male'); (iii) *conjunctions* (AND), where both the range and categorical conditions are applied simultaneously; and (iv) *disjunctions* (OR) of range and categorical constraints, where either one may be satisfied. Details of the constraint specification and implementation are provided in Appendix E. For these tasks, we report the *constraint violation rate* (Stoian et al., 2024), measuring the fraction of samples that break the enforced restriction, the $\alpha$-*score* (Alaa et al., 2022), which quantifies fidelity, and the *utility* using a downstream XGBoost (Chen & Guestrin, 2016) classifier trained on the synthetic data and evaluated against a holdout test set (additional details in Appendix F). We also provide results using additional metrics in Appendix I.5. All results are averaged across five trials to account for variance.

## 4.2 IMPUTATION RESULTS

Tables 1 and 2 report the MSE and imputation accuracy for continuous and categorical features respectively. For continuous features, HARPOON (ours), DiffPuter, and Remasker are generally the best-performing. While the inference-time methods (DiffPuter and HARPOON) remain robust even under high missingness scenarios, training-time methods, such as Remasker and GAIN, de-

grade sharply under increased missing ratios (e.g., Gesture at 75%), probably because they rely on training-time mask patterns and missing ratios, which limits generalisation at inference. LLM-based generation with GReaT is unreliable, occasionally producing extremely high MSE values (e.g. gesture), indicating their limited effectiveness for tabular tasks. While the causal-learning method, MIRACLE, sometimes achieves low errors, its performance can be unstable, collapsing entirely under high missingness (e.g., Default at 75%). This likely arises because MIRACLE constructs a fresh imputer per task at inference rather than leveraging a pre-trained model, which can fail when insufficient information is available under high missing ratios.

For categorical features, HARPOON often achieves the highest accuracy across nearly all datasets and missingness ratios, outperforming DiffPuter, Remasker, and GReaT. While DiffPuter remains competitive, its RePaint strategy (Lugmayr et al., 2022) of independently noising observed and masked entries can introduce semantic mismatches, reducing alignment with observed categories. Most other methods, except for GReaT, typically only model continuous features, resulting in poor performance on categorical imputation. Overall, we see that HARPOON provides robust and reliable inference-time imputation across both continuous and categorical features.

## 4.3 RESULTS ON INEQUALITY CONSTRAINTS

Table 3 compares the best generative baselines from Tables 1 and 2 (HARPOON, DiffPuter, and GReaT) on general tabular inequality constraints. Deterministic imputers such as Remasker are excluded, as they produce identical outputs when inputs are identical, making them unsuitable for such tasks which do not have partially observable ground-truths.

For range constraints, both DiffPuter and GReaT exhibit high violation rates (74–96%). This likely stems from their approach of treating conditional tasks as standard imputation problems, which is ill-suited to inequality constraints because there is no observed ground truth, only a plausible range. As a result, these methods rely on rejection sampling. In contrast, HARPOON handles these tasks using a differentiable loss specifically designed for inequalities (see eq. 9, Appendix D). This provides greater flexibility, reducing violations to **below 8%** on Adult and Default, and 20% on Shoppers, while generally achieving the fidelity and utility scores.

Table 2: Categorical imputation accuracy ($\uparrow$) for MAR mask.

| Ratio | Method | Adult | Default | Shoppers |
|---|---|---|---|---|
| 0.25 | GAIN | $22.11_{6.97}$ | $51.30_{2.66}$ | $41.25_{7.70}$ |
| | Miracle | $42.10_{5.85}$ | $58.51_{5.05}$ | $27.22_{6.83}$ |
| | GReaT | $22.88_{1.56}$ | $44.13_{0.49}$ | $37.69_{5.77}$ |
| | Remasker | $46.81_{11.19}$ | $69.63_{1.76}$ | $39.83_{6.07}$ |
| | DiffPuter | $\mathbf{69.47}_{3.05}$ | $\underline{71.58}_{2.53}$ | $\mathbf{56.19}_{4.49}$ |
| | HARPOON | $\underline{69.44}_{3.11}$ | $\mathbf{73.49}_{2.53}$ | $\underline{51.17}_{5.20}$ |
| 0.50 | GAIN | $21.85_{8.01}$ | $42.28_{3.24}$ | $27.06_{8.02}$ |
| | Miracle | $33.73_{8.89}$ | $55.11_{1.93}$ | $30.09_{5.93}$ |
| | GReaT | $45.97_{2.50}$ | $55.81_{1.97}$ | $37.12_{5.95}$ |
| | Remasker | $45.16_{11.05}$ | $\underline{63.00}_{1.27}$ | $38.44_{6.27}$ |
| | DiffPuter | $\underline{63.33}_{4.58}$ | $62.92_{3.06}$ | $\mathbf{52.14}_{4.67}$ |
| | HARPOON | $\mathbf{64.80}_{3.29}$ | $\mathbf{69.55}_{1.91}$ | $\underline{48.39}_{5.00}$ |
| 0.75 | GAIN | $22.41_{1.45}$ | $20.69_{1.26}$ | $17.61_{4.59}$ |
| | Miracle | $35.46_{2.51}$ | $41.74_{14.87}$ | $38.33_{4.68}$ |
| | GReaT | $50.16_{3.07}$ | $\underline{56.93}_{1.43}$ | $37.89_{4.35}$ |
| | Remasker | $38.67_{1.43}$ | $50.28_{1.18}$ | $44.15_{5.79}$ |
| | DiffPuter | $\underline{56.99}_{3.31}$ | $49.24_{3.40}$ | $\mathbf{50.94}_{5.07}$ |
| | HARPOON | $\mathbf{58.81}_{3.85}$ | $\mathbf{61.48}_{3.83}$ | $\underline{48.37}_{4.39}$ |

For categorical constraints, all methods achieve **near-zero** violations. However, HARPOON generally attains the highest precision, despite a slightly higher violation rate than the others. This is because both DiffPuter and GReaT handle categorical conditions like imputation tasks and enforce ground-truth conditions (e.g., *color* = 'red') using hard substitutions, which can push samples off-manifold. In contrast, HARPOON enforces constraints through soft differentiable losses, allowing all features to be coherently adapted via gradient steps at the cost of a small violation rate ($\leq \mathbf{2\%}$). Hence, samples remain aligned with the manifold giving outputs with high fidelity and utility scores.

Under conjunctions, DiffPuter and GReaT frequently fail, with violation rates between 78–97%. In contrast, HARPOON maintains low violations (**2–12%**) while maintaining strong fidelity and utility across datasets. This demonstrates its ability to flexibly satisfy diverse types of constraints without sacrificing sample quality. Disjunctions show that HARPOON has lowest violation rates amongst methods, and is also consistently lower than the worst of its range or categorical constraint in isolation. This is intuitive, since the conjunctions increase the feasible region, making it easier to satisfy. However, fidelity scores sometimes show a stark drop (Default, Shoppers). We hypothesise that this is because the model's gradient updates may prioritize one condition over another, skewing the distribution toward certain feasible regions (see Figure 4, Appendix D). Nevertheless, high utility scores indicate that the generated samples remain useful for downstream tasks.

Table 3: Constraint Violation rate ($\downarrow$), fidelity ($\alpha$-score ($\uparrow$)) and downstream utility ($\uparrow$) of methods.

| Constr. | Method | Adult | | | Default | | | Shoppers | | |
|---|---|---|---|---|---|---|---|---|---|---|
| | | Viol. % | $\alpha$-score | Utility | Viol. % | $\alpha$-score | Utility | Viol. % | $\alpha$-score | Utility |
| *Range* | DiffPuter | $\underline{73.86}_{1.03}$ | $\underline{0.83}_{0.02}$ | $\underline{0.94}_{0.00}$ | $\underline{88.53}_{1.21}$ | $0.65_{0.02}$ | $\underline{0.78}_{0.15}$ | $\underline{76.69}_{1.61}$ | $0.68_{0.01}$ | $\mathbf{0.89}_{0.01}$ |
| | GReaT | $83.58_{0.69}$ | $\mathbf{0.92}_{0.01}$ | $\mathbf{0.95}_{0.00}$ | $95.37_{0.83}$ | $\underline{0.67}_{0.01}$ | $0.70_{0.18}$ | $96.53_{0.46}$ | $\mathbf{0.88}_{0.00}$ | $0.35_{0.43}$ |
| | Harpoon | $\mathbf{3.06}_{0.26}$ | $\mathbf{0.92}_{0.01}$ | $\mathbf{0.95}_{0.00}$ | $\mathbf{7.78}_{0.69}$ | $\mathbf{0.79}_{0.02}$ | $\mathbf{0.85}_{0.01}$ | $\mathbf{20.55}_{0.95}$ | $0.60_{0.01}$ | $\mathbf{0.89}_{0.01}$ |
| *Category* | DiffPuter | $\mathbf{0.00}_{0.00}$ | $0.83_{0.02}$ | $0.87_{0.01}$ | $\mathbf{0.00}_{0.00}$ | $\underline{0.85}_{0.02}$ | $0.81_{0.06}$ | $\mathbf{0.00}_{0.00}$ | $0.52_{0.02}$ | $\underline{0.82}_{0.12}$ |
| | GReaT | $\mathbf{0.00}_{0.00}$ | $0.72_{0.02}$ | $0.87_{0.01}$ | $\mathbf{0.00}_{0.00}$ | $\underline{0.85}_{0.01}$ | $\mathbf{0.83}_{0.01}$ | $\mathbf{0.00}_{0.00}$ | $\mathbf{0.86}_{0.03}$ | $0.71_{0.35}$ |
| | Harpoon | $\underline{2.18}_{0.36}$ | $\mathbf{0.86}_{0.02}$ | $\mathbf{0.88}_{0.01}$ | $\underline{0.91}_{0.19}$ | $\mathbf{0.92}_{0.01}$ | $\mathbf{0.83}_{0.00}$ | $\mathbf{0.00}_{0.00}$ | $\underline{0.71}_{0.03}$ | $\mathbf{0.88}_{0.01}$ |
| *Range* AND *Category* | DiffPuter | $\underline{78.88}_{6.41}$ | $0.73_{0.04}$ | $\mathbf{0.92}_{0.03}$ | $\underline{91.30}_{0.87}$ | $0.59_{0.03}$ | $\underline{0.56}_{0.13}$ | $\underline{78.68}_{3.28}$ | $\mathbf{0.76}_{0.02}$ | $\mathbf{0.85}_{0.02}$ |
| | GReaT | $80.22_{3.38}$ | $\mathbf{0.90}_{0.01}$ | $0.91_{0.02}$ | $94.79_{1.77}$ | $\underline{0.61}_{0.02}$ | $0.52_{0.16}$ | $95.23_{1.06}$ | $0.73_{0.02}$ | $\underline{0.84}_{0.01}$ |
| | Harpoon | $\mathbf{3.37}_{1.42}$ | $0.82_{0.05}$ | $\underline{0.91}_{0.03}$ | $\mathbf{2.98}_{0.77}$ | $\mathbf{0.87}_{0.01}$ | $\mathbf{0.81}_{0.02}$ | $\mathbf{12.05}_{2.19}$ | $0.81_{0.04}$ | $\mathbf{0.85}_{0.01}$ |
| *Range* OR *Category* | DiffPuter | $\underline{63.22}_{0.57}$ | $\mathbf{0.88}_{0.00}$ | $\underline{0.93}_{0.00}$ | $\underline{23.03}_{0.26}$ | $\underline{0.88}_{0.01}$ | $\underline{0.82}_{0.07}$ | $\underline{50.56}_{0.42}$ | $\underline{0.53}_{0.01}$ | $\underline{0.88}_{0.01}$ |
| | GReaT | $67.14_{0.31}$ | $\mathbf{0.88}_{0.01}$ | $\mathbf{0.94}_{0.00}$ | $24.04_{0.14}$ | $\mathbf{0.90}_{0.00}$ | $\mathbf{0.84}_{0.01}$ | $65.76_{0.33}$ | $\mathbf{0.82}_{0.02}$ | $0.35_{0.42}$ |
| | Harpoon | $\mathbf{3.04}_{0.07}$ | $\underline{0.87}_{0.01}$ | $\mathbf{0.93}_{0.00}$ | $\mathbf{4.23}_{0.35}$ | $0.65_{0.01}$ | $0.81_{0.00}$ | $\mathbf{20.20}_{1.26}$ | $0.47_{0.01}$ | $\mathbf{0.89}_{0.01}$ |

Table 4: Harpoon's imputation performance (MAR) under different inference-time losses using a model pre-trained on MSE loss.

| Ratio | Loss | Adult | | Default | | Shoppers | |
|---|---|---|---|---|---|---|---|
| | | MSE ($\downarrow$) | Acc. ($\uparrow$) | MSE ($\downarrow$) | Acc. ($\uparrow$) | MSE ($\downarrow$) | Acc. ($\uparrow$) |
| 0.25 | MAE | $\mathbf{0.99}_{0.08}$ | $\mathbf{69.44}_{3.11}$ | $\mathbf{0.51}_{0.06}$ | $\mathbf{73.49}_{2.53}$ | $\mathbf{0.73}_{0.06}$ | $51.17_{5.20}$ |
| | MSE | $1.15_{0.09}$ | $68.39_{3.04}$ | $0.68_{0.10}$ | $71.57_{2.16}$ | $0.86_{0.11}$ | $\mathbf{51.51}_{5.32}$ |
| | MAE + CE | $\underline{1.05}_{0.10}$ | $\underline{69.30}_{3.02}$ | $\underline{0.54}_{0.07}$ | $\underline{73.01}_{2.48}$ | $\underline{0.77}_{0.08}$ | $51.13_{5.35}$ |
| | MSE + CE | $1.31_{0.13}$ | $66.62_{2.90}$ | $0.77_{0.12}$ | $71.39_{2.45}$ | $1.04_{0.16}$ | $\underline{51.27}_{5.28}$ |
| 0.50 | MAE | $\mathbf{1.08}_{0.06}$ | $\mathbf{64.80}_{3.29}$ | $\mathbf{0.53}_{0.06}$ | $\mathbf{69.55}_{1.91}$ | $\mathbf{0.81}_{0.06}$ | $\mathbf{48.39}_{5.00}$ |
| | MSE | $1.19_{0.07}$ | $63.28_{3.07}$ | $0.67_{0.08}$ | $67.60_{1.76}$ | $0.90_{0.10}$ | $\underline{48.24}_{4.92}$ |
| | MAE + CE | $\underline{1.14}_{0.08}$ | $\underline{64.56}_{3.15}$ | $\underline{0.55}_{0.07}$ | $\underline{69.26}_{1.90}$ | $\underline{0.85}_{0.06}$ | $48.05_{4.98}$ |
| | MSE + CE | $1.32_{0.08}$ | $62.17_{2.90}$ | $0.73_{0.09}$ | $67.72_{1.76}$ | $1.07_{0.12}$ | $48.04_{5.06}$ |
| 0.75 | MAE | $\mathbf{1.26}_{0.07}$ | $\mathbf{58.81}_{3.85}$ | $\mathbf{0.72}_{0.06}$ | $\mathbf{61.48}_{3.83}$ | $\mathbf{1.10}_{0.05}$ | $\mathbf{48.37}_{4.39}$ |
| | MSE | $1.38_{0.06}$ | $57.18_{3.41}$ | $0.88_{0.07}$ | $58.81_{3.93}$ | $1.17_{0.05}$ | $47.76_{4.40}$ |
| | MAE + CE | $\underline{1.30}_{0.06}$ | $\underline{58.73}_{3.83}$ | $\underline{0.73}_{0.07}$ | $\underline{61.38}_{3.85}$ | $\underline{1.15}_{0.04}$ | $\underline{48.12}_{4.44}$ |
| | MSE + CE | $1.44_{0.04}$ | $57.20_{3.39}$ | $0.91_{0.09}$ | $59.63_{4.11}$ | $1.29_{0.05}$ | $47.68_{4.56}$ |

## 4.4 ABLATION: CHOICE OF INFERENCE-TIME LOSS

Table 4 compares the imputation scores of various inference-time loss functions for HARPOON. Across all scenarios, the best performance, both in MSE and imputation accuracy scores, consistently comes from MAE or MAE with cross-entropy (CE). In contrast, the MSE loss variants perform worse, despite MSE being the training-time objective. This is because MAE is sparsity-inducing, which aligns well with the structure of discrete features represented as sparse one-hot vectors, while MSE "spreads" errors across dimensions, giving imprecise categorical predictions.

*Remark* 2. The ablation results in Table 4 have important implications. It shows that using a tabular-specific loss at inference (e.g. sparsity-inducing MAE) can even outperform the loss used in training, directly verifying Theorem 3.2. Once the denoiser learns to project orthogonally onto the data manifold, the gradients of *any* differentiable inference-time objective lie in the tangent space of the manifold, enabling guidance with respect to arbitrary loss choices. Notably, this does not require special handling of discrete features during training or separate discrete and continuous diffusion processes, like the ones introduced in Sec. 2.

## 4.5 ABLATION: CHOICE OF GUIDANCE SCHEDULE

Motivated by the discussions in Sec.3.3 and the observations in Figure 3, Harpoon injects tangential corrections at *every* denoising step during inference, since the gradients remain nearly orthogonal for the entire denoising process. But Theorem 3.2, says that orthogonality strictly holds only as outputs approach the manifold $\mathcal{M}_0$. Therefore, we compare two strategies: a *linear schedule*, which

Table 5: Linear vs. fixed guidance strength across denoising steps (MAR).

| Ratio | Type | Adult | | Default | | Shoppers | |
|---|---|---|---|---|---|---|---|
| | | MSE ($\downarrow$) | Acc. ($\uparrow$) | MSE ($\downarrow$) | Acc. ($\uparrow$) | MSE ($\downarrow$) | Acc. ($\uparrow$) |
| 0.25 | Linear | $\mathbf{0.99}_{0.08}$ | $68.91_{2.92}$ | $\mathbf{0.51}_{0.06}$ | $73.21_{2.30}$ | $\mathbf{0.68}_{0.07}$ | $50.70_{5.64}$ |
| | Fixed | $\mathbf{0.99}_{0.08}$ | $\mathbf{69.44}_{3.11}$ | $\mathbf{0.51}_{0.06}$ | $\mathbf{73.49}_{2.53}$ | $0.73_{0.06}$ | $\mathbf{51.17}_{5.20}$ |
| 0.50 | Linear | $1.09_{0.07}$ | $63.46_{3.19}$ | $0.54_{0.05}$ | $67.16_{2.33}$ | $\mathbf{0.80}_{0.08}$ | $47.30_{5.33}$ |
| | Fixed | $\mathbf{1.08}_{0.06}$ | $\mathbf{64.80}_{3.29}$ | $\mathbf{0.53}_{0.06}$ | $\mathbf{69.55}_{1.91}$ | $0.81_{0.06}$ | $\mathbf{48.39}_{5.00}$ |
| 0.75 | Linear | $1.37_{0.07}$ | $55.49_{3.39}$ | $0.78_{0.07}$ | $54.64_{4.29}$ | $1.16_{0.07}$ | $46.70_{4.76}$ |
| | Fixed | $\mathbf{1.26}_{0.07}$ | $\mathbf{58.81}_{3.85}$ | $\mathbf{0.72}_{0.06}$ | $\mathbf{61.48}_{3.83}$ | $\mathbf{1.10}_{0.05}$ | $\mathbf{48.37}_{4.39}$ |

gradually increases the guidance strength ($\eta$) up to a maximum of 0.2 during denoising, and our default *constant schedule*, which maintains a guidance strength of 0.2 for all steps (Table 5). Across all scenarios, the linear schedule consistently leads to lower imputation accuracy and higher MSE. We hypothesize that this occurs because even approximate guidance early in the denoising process helps steer the model toward feasible regions. Delaying guidance until later steps may leave the model insufficient time to converge to regions consistent with the constraints.

### 4.6 RUNTIME COMPARISONS

Table 6 compares the runtimes of various methods on three datasets (more datasets in Appendix I.4) using a 12-core AMD Ryzen 9 5900 processor with an NVIDIA RTX 3090 GPU, CUDA 12.1, Ubuntu 20.04.6, and Pytorch 2.2.1 with Python 3.12. We observe that HARPOON has a runtime within roughly 2× of its "unconditional" proxy, DiffPuter. This is expected because DiffPuter performs only the forward pass and does not propagate gradients, whereas HARPOON interleaves denoising steps with tangential gradient updates, which doubles the function calls to the backbone network. Nevertheless, HARPOON remains extremely efficient, taking **less than 5 seconds** even on the largest dataset, Adult. In contrast, the LLM autoregressive paradigm (GReaT) and the causal graph-based method (Miracle) are

Table 6: Sampling runtimes (seconds) of methods (MAR, 0.25 mask ratio).

| Method | Adult | Default | Gesture |
|---|---|---|---|
| GAIN | $0.14_{0.26}$ | $0.15_{0.28}$ | $0.14_{0.27}$ |
| Miracle | $33.03_{0.99}$ | $44.54_{0.29}$ | $23.09_{0.13}$ |
| GReaT | $65.47_{2.57}$ | $202.71_{7.44}$ | $131.72_{1.96}$ |
| Remasker | $2.68_{5.24}$ | $0.33_{0.53}$ | $0.30_{0.54}$ |
| DiffPuter | $2.83_{0.27}$ | $2.62_{0.26}$ | $0.98_{0.27}$ |
| Harpoon | $4.43_{0.34}$ | $4.05_{0.34}$ | $1.54_{0.35}$ |

significantly slower and have lower imputation quality. The GAN-based method, GAIN, is very fast, but produces poor-quality generations compared to diffusion-based approaches.

## 5 CONCLUSION

We propose HARPOON, a tabular diffusion method for guiding generation to meet diverse constraints. With a geometric perspective on tabular diffusion, we prove that denoisers act as orthogonal projectors, guaranteeing the tangent-space behaviour of inference-time gradients. This allows training models with simple losses, while allowing for task-specific objectives to be applied at inference.

HARPOON'S unlocks practical benefits for real-world tabular modelling. It handles imputation, inequality constraints, and heterogeneous feature types without special-case training procedures. Its manifold-guided updates provide a general guidance mechanism for enforcing constraints at inference, making generation robust and adaptable across diverse tasks. We empirically validate our theoretical findings and show HARPOON'S adaptability in handling diverse constraints.

**Limitations and Future Work.** We model tabular diffusion as a continuous process to fit existing manifold theories. However, extending the theory itself to discrete noising mechanisms (Kotelnikov et al., 2023; Shi et al., 2025) would broaden applicability to other diffusion designs. Beyond tables, applying inference-time guidance to other modalities is a natural next step for improvement.

Overall, our work introduces a new design principle for conditional tabular diffusion, bridging the gap between theory and practice: *keep training simple, scale at inference*.

**Acknowledgements.** This research is partly funded by Priv-GSyn project, 200021E_229204 of Swiss National Science Foundation, the DEPMAT project, P20-22 / N21022, of the research programme Perspectief of the Dutch Research Council, and ASM international NV. This publication was also supported by the Dutch Research Council (VI.Veni.222.439), the Smart Networks and Services Joint Undertaking (SNS JU) under the European Union's Horizon Europe Research and Innovation programme (Grant Agreement No. 101192750)

**Reproducibility Statement.** We provide full implementation details, including model architectures, training hyperparameters, and evaluation protocols in the Appendix. All experiments are run with fixed random seeds and repeated across five independent trials to report mean and standard deviation. Public datasets are used for all experiments, and we open-source our anonymised code: `https://github.com/adis98/Harpoon`. We additionally also provide the full proofs for our theoretical contributions in the Appendix.

**Ethics Statement.** This work advances theoretical and methodological foundations for conditional tabular generation. By enabling flexible inference-time guidance, our framework can support applications where respecting domain constraints is essential, such as enforcing physical laws in scientific simulations, adhering to medical ranges in healthcare analytics, or satisfying financial regulations in risk modelling. All experiments are conducted on standard, publicly available benchmark datasets, and our approach is intended to be a building block for reliable, constraint-aware generative modelling in practical domains. Regarding AI use, Large language models were used to assist us with writing and editing, but all the ideas, designs, and analyses are our own.

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

## A  NOTATIONS

Table 7: Summary of key notations.

| Symbol | Explanation |
|---|---|
| **(A) Data & Manifold Setup** | |
| $d$ | Ambient space dimension |
| $x_0 \in \mathbb{R}^d$ | Clean (un-noised) sample |
| $p(x_0)$ | Clean data distribution |
| $\mathcal{M}_0 \subset \mathbb{R}^d$ | Smooth, connected data manifold |
| $n$ | Intrinsic dimension of $\mathcal{M}_0$ with $n \ll d$ |
| $\varphi : \mathbb{R}^n \to \mathbb{R}^d$ | Smooth manifold embedding function (Assump. 1) |
| $T_{x_0}\mathcal{M}_0$ | Tangent space of $\mathcal{M}_0$ at $x_0$ |
| $B_r(x_0) = \{x : \|x - x_0\|_2 < r\}$ | Local hypersphere of radius $r$ about $x_0$ (Assump. 2) |
| **(B) Forward Diffusion & Noisy Shells** | |
| $t \in \{1, \dots, T\}$ | Diffusion step index; $T$ total steps |
| $\alpha_t$ | Noise schedule at step $t$ |
| $\bar{\alpha}_t = \prod_{s=1}^{t} \alpha_s$ | Cumulative noise schedule |
| $x_t$ | Noisy sample at diffusion step $t$ |
| $\epsilon$ | Sampled white noise |
| $\mathcal{M}_t$ | Noisy shell manifold at step $t$ (Prop. 1) |
| **(C) Denoiser & Training Objective** | |
| $\epsilon_\theta(x_t, t)$ | Denoiser's noise estimate for sample at step $t$ |
| $\mathcal{L}_{\text{train}}$ | MSE training loss eq. 2 |
| **(D) Projection & Dirty Estimate** | |
| $Q_t(.)$ | Dirty estimate mapping function (Def. 1) |
| $\hat{x}_0 = Q_t(.)$ | Dirty estimate point on $\mathcal{M}_0$ |
| $\pi(x_t)$ | Orthogonal projection of $x_t$ onto $\mathcal{M}_0$ (Thm. 3.1) |
| **(E) Guidance & Tangency** | |
| $c \in \mathcal{C}$ | Arbitrary conditioning information |
| $\mathcal{L}_{\text{inf}}(.)$ | Arbitrary differentiable inference-time loss |
| **(F) HARPOON Sampling (Alg. 1)** | |
| $z$ | Sampled noise for stochastic denoising |
| $\sigma_t$ | Noise scale for denoising step |
| $\eta$ | Guidance step size (strength) |
| $x'_{t-1}$ | Unconditional one-step denoised estimate |

## B  PROOFS

**Theorem 3.1** (Limiting behaviour of dirty estimates) *Let $\epsilon_\theta$ be a denoiser trained to predict Gaussian noise under the framework of eqs. 1 and 2, for a clean sample $x_0$. Then, for the mapping $Q_t$ defined in Definition 1, we have $\lim_{\bar{\alpha}_t \to 1} Q_t(x_t) = \pi(x_t)$, where $\pi(x_t)$ denotes the orthogonal projection of $x_t$ onto the data manifold $\mathcal{M}_0$.*

**Proof.** Consider the training objective for a denoiser with parameters $\theta$ under a continuous Gaussian noising framework equation 2:

$$\mathcal{L}_{\text{train}} = \mathbb{E}_{x_0 \sim p(x_0), t \sim U[1,T], \epsilon \sim \mathcal{N}(0,I)} \left[ \|\epsilon_\theta(x_t, t) - \epsilon\|_2^2 \right].$$

Substituting $\epsilon$ from equation 1, we get:

$$\mathcal{L}_{\text{train}} = \mathbb{E}_{x_0, t, \epsilon} \left[ \left\| \epsilon_\theta(x_t, t) - \frac{x_t - \sqrt{\bar{\alpha}_t} x_0}{\sqrt{1 - \bar{\alpha}_t}} \right\|_2^2 \right].$$

$$\mathcal{L}_{\text{train}} = \int_{x_0} \int_{x_t} \int_t \left\| \epsilon_\theta(x_t, t) - \frac{x_t - \sqrt{\bar{\alpha}_t} x_0}{\sqrt{1 - \bar{\alpha}_t}} \right\|_2^2 p(x_t|x_0, t) p(x_0) p(t) dt dx_t dx_0$$

For fixed $x_t$ and $t$, $\epsilon_\theta(x_t, t)$ is optimized by minimizing the loss, i.e., setting $\nabla_{\epsilon_\theta(x_t, t)} \mathcal{L}_{\text{train}} = 0$:

$$\nabla_{\epsilon_\theta(x_t, t)} \mathcal{L}_{\text{train}} = \int_{x_0} 2 \left( \epsilon_\theta(x_t, t) - \frac{x_t - \sqrt{\bar{\alpha}_t} x_0}{\sqrt{1 - \bar{\alpha}_t}} \right) p(x_t|x_0, t) p(x_0) dx_0 = 0$$

$$\epsilon_\theta(x_t, t) \int_{x_0} p(x_t|x_0, t) p(x_0) dx_0 = \int_{x_0} \left( \frac{x_t - \sqrt{\bar{\alpha}_t} x_0}{\sqrt{1 - \bar{\alpha}_t}} \right) p(x_t|x_0, t) p(x_0) dx_0$$

Dividing both sides by $\int_{x_0} p(x_t|x_0, t) p(x_0) dx_0$, we get:

$$\epsilon_\theta(x_t, t) = \frac{x_t}{\sqrt{1 - \bar{\alpha}_t}} - \frac{\sqrt{\bar{\alpha}_t}}{\sqrt{1 - \bar{\alpha}_t}} \frac{\int_{x_0} x_0 p(x_t|x_0, t) p(x_0) dx_0}{\int_{x_0} p(x_t|x_0, t) p(x_0) dx_0}$$

Applying Bayes theorem, we can rewrite this as follows:

$$\epsilon_\theta(x_t, t) = \frac{x_t}{\sqrt{1 - \bar{\alpha}_t}} - \frac{\sqrt{\bar{\alpha}_t} F(x_t, t)}{\sqrt{1 - \bar{\alpha}_t}}, \quad \text{where} \quad F(x_t, t) = \frac{\int_{x_0} x_0 p(x_0|x_t, t) p(\cancel{x_t}) dx_0}{\int_{x_0} p(x_0|x_t, t) p(\cancel{x_t}) dx_0} \quad (4)$$

We partition $F(x_t, t)$ into contributions from points $x_0$ in a region $E \subset \mathcal{M}_0$, chosen inside a ball of radius $\varepsilon \in (0, \|x_t - \pi(x_t)\|_2)$ around the orthogonal projection $\pi(x_t)$, and its complement $\mathcal{M}_0 \setminus E$ (Stanczuk et al., 2024). Under the Gaussian noising framework, the conditional distribution satisfies $p(x_0|x_t, t) \sim \mathcal{N}(x_0|x_t, \sigma_t^2 I)$.

$$F(x_t, t) = \frac{\int_E x_0 \mathcal{N}(x_0|x_t, \sigma_t^2 I) \, dx_0}{\int_{\mathcal{M}_0} \mathcal{N}(x_0|x_t, \sigma_t^2 I) \, dx_0} + \frac{\int_{\mathcal{M}_0 \setminus E} x_0 \mathcal{N}(x_0|x_t, \sigma_t^2 I) \, dx_0}{\int_{\mathcal{M}_0} \mathcal{N}(x_0|x_t, \sigma_t^2 I) \, dx_0}$$

We then divide both the numerator and denominator by $\int_E \mathcal{N}(x_0|x_t, \sigma_t^2 I) \, dx_0$:

$$F(x_t, t) = \frac{\frac{\int_E x_0 \cdot \mathcal{N}(x_0|x_t, \sigma_t^2 I) \, dx_0}{\int_E \mathcal{N}(x_0|x_t, \sigma_t^2 I) \, dx_0} + \frac{\int_{\mathcal{M}_0 \setminus E} x_0 \cdot \mathcal{N}(x_0|x_t, \sigma_t^2 I) \, dx_0}{\int_E \mathcal{N}(x_0|x_t, \sigma_t^2 I) \, dx_0}}{\frac{\int_E \mathcal{N}(x_0|x_t, \sigma_t^2 I) \, dx_0}{\int_E \mathcal{N}(x_0|x_t, \sigma_t^2 I) \, dx_0} + \frac{\int_{\mathcal{M}_0 \setminus E} \mathcal{N}(x_0|x_t, \sigma_t^2 I) \, dx_0}{\int_E \mathcal{N}(x_0|x_t, \sigma_t^2 I) \, dx_0}}.$$

Applying **Lemma D.9** (Stanczuk et al., 2024), for a general Gaussian distribution, the contributions from $\mathcal{M}_0 \setminus E$ become negligibly small as $\sigma_t \to 0$. This simplifies the expression to:

$$\lim_{\sigma_t \to 0} F(x_t, t) = \frac{\int_E x_0 \cdot \mathcal{N}(x_0|x_t, \sigma_t^2 I) \, dx_0}{\int_E \mathcal{N}(x_0|x_t, \sigma_t^2 I) \, dx_0}.$$

Subtracting $\pi(x)$ from both sides, we get:

$$\lim_{\sigma_t \to 0} F(x_t, t) - \pi(x_t) = \frac{\int_E (x_0 - \pi(x_t)) \cdot \mathcal{N}(x_0|x_t, \sigma_t^2 I) \, dx_0}{\int_E \mathcal{N}(x_0|x_t, \sigma_t^2 I) \, dx_0}.$$

Since $\|x_0 - \pi(x)\| \le \varepsilon$ for all $x_0 \in E$, we can bound the difference:

$$0 \le \|F(x_t, t) - \pi(x_t)\| \le \varepsilon \cdot \frac{\cancel{\int_E \mathcal{N}(x_0|x_t, \sigma_t^2 I) \, dx_0}}{\cancel{\int_E \mathcal{N}(x_0|x_t, \sigma_t^2 I) \, dx_0}}.$$

As $\varepsilon$ is a user-defined parameter that can be set arbitrarily small when constructing $E$, we get:

$$\lim_{\sigma_t \to 0} F(x_t, t) = \pi(x_t).$$

Upon substituting this result for $F(x_t, t)$ in equation 4, we get:

$$\lim_{\sigma_t \to 0} \epsilon_\theta(x_t, t) = \frac{x_t}{\sqrt{1 - \bar{\alpha}_t}} - \frac{\sqrt{\bar{\alpha}_t}}{\sqrt{1 - \bar{\alpha}_t}} \pi(x_t)$$

Since $\sigma_t^2 = 1 - \bar{\alpha}_t$, we have:

$$\lim_{\bar{\alpha}_t \to 1} Q_t(x_t) = \frac{x_t}{\sqrt{\bar{\alpha}_t}} - \frac{\sqrt{1 - \bar{\alpha}_t} \epsilon_\theta(x_t, t)}{\sqrt{\bar{\alpha}_t}} = \pi(x_t)$$

Hence $Q_t(x_t)$ points to the orthogonal projection on the data manifold $\mathcal{M}_0$.

**Key difference from Chung et al. (2022):** While our result is similar to Proposition 2 from Chung et al. (2022), the key difference lies in how we treat the integral in equation 4. Chung et al. (2022) assume this integral behaves as a weighted average over all points on the manifold, relying on the symmetry of the normal distribution to argue that the denoised estimate points to the orthogonal projection. However, this assumption requires the manifold to be globally linear, not just locally linear, since the integral spans the entire manifold. In contrast, our proof makes no such assumption on the manifold's geometry, generalizing the result to non-linear manifolds.

**Theorem 3.2** (Manifold-constrained inference-time gradients) *Let $c$ denote arbitrary conditioning information and assume $Q_t$ from Definition 1 acts as an orthogonal projection onto the data manifold $\mathcal{M}_0$ at $\hat{x}_0 = Q_t(x_t)$. Then, for any differentiable inference-time loss function $\mathcal{L}_{\inf} : \mathbb{R}^d \times \mathcal{C} \to \mathbb{R}$, defined by $\mathcal{L}_{\inf}(\hat{x}_0; c)$, the gradient lies in the tangent space of $\mathcal{M}_0$ at $\hat{x}_0$, i.e., $\nabla_{x_t} \mathcal{L}_{\inf}(\hat{x}_0, c) \in T_{\hat{x}_0} \mathcal{M}_0$.*

**Proof.** Let $v \in \mathbb{R}^d$ be an arbitrary vector. We decompose it as $v = v_T + v_N$, where $v_T \in T_{Q_t(x_t)} \mathcal{M}_0$ is the tangential component and $v_N \perp T_{Q_t(x_t)} \mathcal{M}_0$ is the normal component.

By Assumption 2, the local behaviour of $Q_t$ satisfies the following:

$$\lim_{s \to 0} Q_t(x_t + sv) = \lim_{s \to 0} (Q_t(x_t) + sv_T)$$

Taking the partial derivative with respect to $s$ and evaluating at $s = 0$ gives the following:

$$\frac{\partial}{\partial s} Q_t(x_t + sv) \Big|_{s=0} = \frac{\partial}{\partial s} (Q_t(x_t) + sv_T) \Big|_{s=0}$$

The left-hand side is $J_{Q_t} v$ (where $J_{Q_t}$ is the Jacobian), so the expression evaluates to the following:

$$J_{Q_t} v = v_T. \tag{5}$$

Now, apply the chain rule for a differentiable loss $\mathcal{L}_{\inf}(\hat{x}_0, c)$:

$$\nabla_{x_t} \mathcal{L}_{\inf}(\hat{x}_0, c) = \frac{\partial \mathcal{L}_{\inf}(\hat{x}_0, c)}{\partial x_t} = \frac{\partial \mathcal{L}_{\inf}(Q_t(x_t), c)}{\partial x_t}$$

$$\nabla_{x_t} \mathcal{L}_{\inf}(\hat{x}_0, c) = J_{Q_t}^T \times \frac{\partial \mathcal{L}_{\inf}}{\partial \hat{x}_0}$$

$$\nabla_{x_t} \mathcal{L}_{\inf}(\hat{x}_0, c) = J_{Q_t}^T w \quad (\text{since } \frac{\partial \mathcal{L}_{\inf}}{\partial \hat{x}_0} = \text{some } w \in \mathbb{R}^d) \tag{6}$$

From equation 5, we have:

$$v^T J_{Q_t}^T = v_T^T$$
$$\therefore v^T J_{Q_t}^T w = v_T^T w$$
$$v_N^T J_{Q_t}^T w + v_T^T J_{Q_t}^T w = v_T^T w$$
$$v_N^T J_{Q_t}^T w + v_T^T J_{Q_t}^T w = v_T^T w_T + v_T^T w_N$$

The term $v_T^T w_N = 0$ since the vectors are perpendicular to one another, yielding:

$$v_N^T J_{Q_t}^T w + v_T^T J_{Q_t}^T w = v_T^T w_T$$

Upon matching terms, since $v_N$ and $v_T$ are perpendicular to one another, we have the following:

$$J_{Q_t}^T w = w_T$$

Substituting this in equation 6, we get the final result:

$$\nabla_{x_t} \mathcal{L}_{\inf}(\hat{x}_0, c) = w_T \in T_{Q_t(x_t)} \mathcal{M}_0$$

# C DATASETS

We used eight widely adopted benchmark datasets: **Adult**[2], **Default** of Credit Card Clients[3], **Magic** Gamma Telescope[4], Online **Shoppers** Purchasing Intention[5], **Gesture** Phase Segmentation[6], **Letter** Recognition[7], and Dry **Bean**[8]. Additionally, we included the **California Housing** dataset from Kaggle[9]. The statistics of these datasets are presented in Table 8, each of which is partitioned using a random 70-30 train-test split. Prior to splitting the datasets, we filtered out the rows having missing values, similar to Zhang et al. (2025). We also remove the target column from training and evaluation to avoid leaking label-feature correlations, following Zhang et al. (2025).

Table 8: Statistics of datasets. # Num stands for the number of numerical columns, and # Cat stands for the number of categorical columns.

| Dataset | # Rows | # Num | # Cat | # Train (In-sample) | # Test (Out-of-Sample) |
|---|---|---|---|---|---|
| **California** Housing | 20,640 | 9 | – | 14,447 | 6,193 |
| **Letter** Recognition | 20,000 | 16 | – | 14,000 | 6,000 |
| **Gesture** Phase Segmentation | 9,522 | 32 | – | 6,665 | 2,857 |
| **Magic** Gamma Telescope | 19,020 | 10 | – | 13,314 | 5,706 |
| Dry **Bean** | 13,610 | 17 | – | 9,527 | 4,083 |
| **Adult** Income | 32,561 | 6 | 8 | 22,792 | 9,769 |
| **Default** of Credit Card Clients | 30,000 | 14 | 10 | 21,000 | 9,000 |
| Online **Shoppers** Purchase | 12,330 | 10 | 7 | 8,631 | 3,699 |

# D TABULAR CONDITIONS: IMPUTATION AND INEQUALITY CONSTRAINTS

While the manifold assumptions describe the geometry of noisy data, our ultimate goal is to do *conditional tabular generation*. Such conditions arise in several forms:

**Imputation under masks.** A binary mask $m \in \{0, 1\}^d$ specifies observed ($m = 0$) and missing/masked ($m = 1$) entries for a ground-truth sample $x_0 \in \mathcal{M}_0$. The conditional target is given by $(1 - m) \odot x_0$, i.e., the subset of ground truth values to reconstruct. We can measure the likelihood of a denoised estimate ($\hat{x}_0$) adhering to the conditions through a reconstruction loss such as

$$\mathcal{L}_{\text{imp}}(\hat{x}_0, x_0, m) = \|(1 - m) \odot (\hat{x}_0 - x_0)\|_p, \tag{7}$$

for $p \in \{1, 2\}$, corresponding to the Mean Absolute Error (MAE) or MSE, respectively.

**Inequality constraints.** Beyond imputation tasks, conditions can also impose inequality constraints:

$$\min_{\hat{x}_0 \in \mathbb{R}^d} f(\hat{x}_0) \quad \text{s.t.} \quad \ell \le g(\hat{x}_0) \le u, \quad h(\hat{x}_0) = v, \tag{8}$$

where $\ell, u$ are elementwise lower and upper bounds (e.g., `Age` $\in [0, 120]$, `Salary` $\ge 0$), $g(\cdot)$ allows transformations or subset selection. The function $h(\cdot)$ could be a selector that enforces features to match a particular discrete or continuous value $v$. Feasibility can be approximated using soft penalties as proposed by (Donti et al., 2021), which we can write in a more general form as follows:

$$\mathcal{L}_{\text{soft}}(\hat{x}_0) = f(\hat{x}_0) + \lambda_g \Big( \|\text{ReLU}(\ell - g(\hat{x}_0))\|_i + \|\text{ReLU}(g(\hat{x}_0) - u)\|_j \Big) + \lambda_h \|h(\hat{x}_0) - v\|_k, \tag{9}$$

where $i, j, k$ are suitably chosen norms (e.g., 1, 2, or $\infty$). While useful in principle, the norm is to incorporate these constraints directly as a training-time objective for a model (Donti et al.,

---

[2] `https://archive.ics.uci.edu/static/public/2/adult.zip`
[3] `https://archive.ics.uci.edu/static/public/350/default+of+credit+card+clients.zip`
[4] `https://archive.ics.uci.edu/static/public/159/magic+gamma+telescope.zip`
[5] `https://archive.ics.uci.edu/static/public/468/online+shoppers+purchasing+intention+dataset.zip`
[6] `https://archive.ics.uci.edu/static/public/302/gesture+phase+segmentation.zip`
[7] `https://archive.ics.uci.edu/static/public/59/letter+recognition.zip`
[8] `https://archive.ics.uci.edu/static/public/602/dry+bean+dataset.zip`
[9] `https://www.kaggle.com/datasets/camnugent/california-housing-prices`

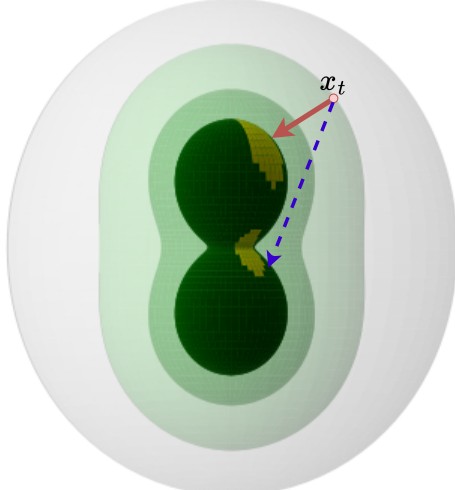

Figure 4: Disconnected submanifolds under disjunctive constraints, e.g. (*colour*=red OR *colour*=blue). Point $x_t$ likely favours the region indicated by red arrow due to the proximity and larger feasible area imposed by the constraint.

2021). This practice presents two limitations: (i) some constraints may correspond to rare events or infrequent classes, making it difficult to get enough samples for reliable training, and (ii) constraints may change at inference (e.g., Age $> 30$ replaced by Age $< 15$), necessitating retraining.

We see that different conditional tasks naturally correspond to different optimisation objectives at inference time, motivating the central question of our work.

**Conjunctions and disjunctions of conditions.** Equation 9 naturally handles conjunctions (AND) of multiple constraints since the overall loss drops to zero when all individual constraints are satisfied. To handle disjunctions (OR), we modify the loss by taking the *product* of individual constraint violations. For instance, for a disjunction such as $\hat{x}_0 > a$ or $\hat{x}_0 < b$, we can formulate it as

$$\mathcal{L}_{\text{disj}}(\hat{x}_0) = ||\text{ReLU}(a - \hat{x}_0)|| \cdot ||\text{ReLU}(\hat{x}_0 - b)||, \tag{10}$$

which drops to zero if either condition is satisfied.

Interestingly, disjunctive constraints can correspond to *disconnected sub-manifolds* of the feasible region. As Figure 4 illustrates, the model may end up favouring one condition over the other depending on which feasible region covers a larger portion of the data manifold, or which region the noisy point $x_t$ is closer to during denoising.

# E    EXPERIMENTAL CONFIGURATION OF TABULAR CONDITIONAL TASKS

We describe the setup of the conditional tasks used in our experiments. We first detail the imputation tasks under different missingness mechanisms, followed by the design of inequality constraints.

## E.1    IMPUTATION TASKS

For imputation, we simulate missingness patterns under three standard assumptions:

**MCAR (Missing Completely at Random).** Entries are removed uniformly at random, independent of the observed or unobserved values. We implement this by masking a random subset of entries at a given missingness ratio, following Zhang et al. (2025).

**MAR (Missing at Random).** Entries are removed conditionally on observed variables. This ensures missingness depends on observed data but not on the missing entries themselves. We implement MAR scenario following the approach in Zhang et al. (2025); Zhao et al. (2023). Specifically, we keep aside a number of columns as the fixed input features and train a logistic model on these features to generate the missing values for the rest. We follow line of search to tune the bias term to get the desired degree of missingness.

**MNAR (Missing Not at Random).** Entries are removed in a way that depends on their own (unobserved) values. Following Zhang et al. (2025), we implement this by separating the input features into two groups and generating the missing values for the second group using the logistic model, similar to the MAR tasks. We then mask out values from the first group following the MCAR setting, creating tasks that are more challenging than both MAR and MCAR.

### E.2 INEQUALITY CONSTRAINTS

In addition to imputation, we evaluate conditional generation under inequality constraints. We consider three types of conditions: range, categorical, and their combination.

**Range Constraints.** We first select a continuous feature at random and restrict it to a region in the tail of its empirical distribution, i.e., a region with relatively few samples in the training data. This design ensures that the constraint is non-trivial and represents a realistic stress test for the model. For evaluation, we construct a "ground truth" reference set by applying the same filter on the test set. We then generate an equal number of samples from the model under the specified constraint. Performance is assessed by measuring (i) the fraction of generated rows that violate the range constraint, (ii) downstream utility using an XGBoost classifier, and (iii) the $\alpha$-score (Alaa et al., 2022) of the generated samples, quantifying their fidelity.

**Categorical Constraints.** For categorical constraints, we fix the value of a randomly chosen discrete feature to a particular class, deliberately avoiding the majority class from the training data. As with range constraints, we construct a ground truth by filtering the test set for rows that satisfy the categorical restriction, and we generate an equal number of samples under this condition. Evaluation follows the same protocol: violation rate and alpha-precision relative to the ground truth.

**Conjunctive Constraints.** We also consider conjunctions, where we consider tasks where both range and categorical constraints are applied jointly. In this case, any generated row that violates at least one of the specified constraints is counted as a violation. Ground truth samples are constructed by filtering the test set with both conditions applied simultaneously.

**Disjunctive Constraints.** Finally, we evaluate disjunctive (OR) constraints, where a generated row only needs to satisfy *at least one* of multiple specified conditions. For example, a constraint may require that either a continuous feature $A$ exceeds a threshold $a$, or a categorical feature $B$ takes a particular value $b$. Ground truth samples are constructed by filtering the test set for rows that satisfy *any* of the specified conditions. Such constraints can result in disconnected submanifolds, as illustrated in Figure 4, since each constraint likely spans a different feasible region on the data manifold.

The specific values and classes chosen for constraints, along with the proportion of training samples covering them, are provided in Table 9, demonstrating that these setups constitute a challenging stress cases rather than trivial ones.

## F EVALUATION METRICS

In addition to standard imputation metrics such as MSE and accuracy, we assess the quality of generated tabular data using a combination of fidelity, privacy, and utility metrics. We implement all our metrics using the code available at SynthCity: `https://github.com/vanderschaarlab/synthcity`.

**Fidelity** quantifies how well the generated samples match the distribution of the real data. We use the following:

Table 9: Constraint specifications for range, categorical, conjunctive, and disjunctive tasks across datasets, together with the proportion of samples seen during training satisfying each constraint.

| Dataset | Constraint Type | Feature(s) | Restriction | Train Set Proportion (%) |
|---|---|---|---|---|
| Adult | Range | Age | $\geq 50$ | 21.73 |
| | Categorical | Workclass | = State-gov | 3.9 |
| | Conjunctive | Age, Workclass | $\geq 50$; State-gov | 0.87 |
| | Disjunctive | Age, Workclass | $\geq 50$ or State-gov | 24.83 |
| Default | Range | AGE | $\geq 50$ | 8.9 |
| | Categorical | EDUCATION | = High School | 16.37 |
| | Conjunctive | AGE, EDUCATION | $\geq 50$; High School | 3.4 |
| | Disjunctive | AGE, EDUCATION | $\geq 50$ or High School | 21.88 |
| Shoppers | Range | Administrative | $\geq 4$ | 25.36 |
| | Categorical | VisitorType | = New_Visitor | 14.25 |
| | Conjunctive | Administrative, VisitorType | $\geq 4$; New_Visitor | 4.19 |
| | Disjunctive | Administrative, VisitorType | $\geq 4$ or New_Visitor | 35.42 |

- **$\alpha$-score** (Alaa et al., 2022): The core idea is that real data naturally concentrates in a "typical" high-density region, with the remaining $1 - \alpha$ fraction treated as outliers. The method therefore constructs the *minimum-volume $\alpha$-support* of the real distribution, i.e., the smallest region that contains the most typical fraction $\alpha$ of real samples. The score then measures the proportion of generated samples that fall inside this typical region. In our experiments, higher $\alpha$-scores indicate better alignment of generated samples with the high-density core of the real data, reflecting strong sample fidelity.

- **KS score** (Kolmogorov–Smirnov test): The Kolmogorov–Smirnov (KS) score measures the maximum difference between the cumulative distribution functions (CDFs) of real and synthetic samples. Intuitively, it quantifies how closely the synthetic data reproduces the marginal distribution of each feature. In the SynthCity implementation, higher KS scores indicate stronger similarity between the real and synthetic distributions, while lower scores indicate substantial divergence.

- **Detection score**: The detection score measures how easily a classifier can distinguish real data from synthetic data. We train a linear classifier on the pooled dataset containing both real and synthetic samples, using K-fold cross-validation to estimate robustness. The metric reports the average AUC-ROC obtained when predicting whether a sample is real or synthetic. Lower scores (close to zero) indicate that the sets are indistinguishable.

**Privacy** metrics assess the risk of identifying real data points from the synthetic dataset. We use the **Identifiability** metric, which estimates how likely it is that a synthetic sample is "too similar" to a real one, indicating potential privacy leakage. Intuitively, the metric checks whether any synthetic record lies *closer* to a real sample than the distance between real samples with each other. If so, the synthetic sample may inadvertently reveal an individual. Higher scores indicate a greater risk of recovering real data (Yoon et al., 2020).

**Utility** quantifies how useful the generated data is for downstream tasks. We train an XGBoost classifier on the generated dataset and evaluate its accuracy on the real test set. This reflects whether the synthetic data preserves predictive relationships present in the original dataset. In our experiments, we predict the gender, the marital status, and the operating system as the targets, for Adult, Default, and Shoppers respectively.

## G  BASELINES AND IMPLEMENTATION DETAILS

### G.1  NON-DIFFUSION BASELINES

We implement most of the baseline methods according to the publicly available codebases.

- Remasker (Du et al., 2024): `https://github.com/tydusky/remasker`. The framework automatically masks a fraction of values during training (50%), which we leave unchanged.

- MIRACLE (Kyono et al., 2021) and GAIN (Yoon et al., 2018): `https://github.com/vanderschaarlab/hyperimpute`. For GAIN, we updated the plugin code to separate pre-training from inference-time masking. Following the paper (Yoon et al., 2018), we pre-train the model using 20% missing data under an MCAR mask, and evaluate at inference on various masks.
- GReaT (Borisov et al., 2023): `https://pypi.org/project/be-great/`. Since the model sometimes fails to generate all missing in one go, we give it up to five retries to fill in the whole table. If it still fails to generate some values, we replace them with the mean for numerical columns. Categorical columns were fixed using random replacement.

All the above baselines (except for GReaT) require tabular data consisting of purely numerical inputs, so we use integer encoding to represent discrete features as numbers. The train and test splits are then standardized using the mean and standard deviation of the training set. For GReaT, no special encoding is required as the tokenizer automatically creates a suitable representation for the values. For evaluation, MSE scores are reported on the standardized data (for numerical columns) for consistency across all baselines and datasets.

### G.2 DIFFUSION BACKBONES AND HYPERPARAMETERS

For all diffusion-based baselines (HARPOON, DiffPuter (Zhang et al., 2025), we adopt a shared MLP architecture as the backbone for consistency. The network consists of 5 linear layers with swish activations. For the timestep embeddings, we use a 2-layer MLP with 1024 channels for the positional embedding. We use one-hot encoding for the categorical features and we standardize all numerical features similar to the other baselines in Sec. G.1. Evaluation is also performed in a similar manner for consistency across baselines.

The diffusion process is configured with $\alpha_1 = 0.9999$ and $\alpha_T = 0.98$, consistent with prior work (Ho et al., 2020). We use $T = 200$ diffusion steps for both noising and denoising using a batch size of 1024 trained for 1000 epochs. We use a learning rate of $10^{-4}$ and a guidance step size $\eta = 0.2$ (HARPOON only) for all datasets and task configurations. When sampling using HARPOON, we use the MAE loss for imputation tasks. For general inequality constraints (equation 8), we use coefficients $\lambda_g, \lambda_h = 1$, the 2-norm for $i, j$ and the 1-norm for $k$.

## H TRAINING AND SAMPLING ALGORITHMS

### H.1 TRAINING

During training (see Algorithm 2), we adopt the standard diffusion objective. At each iteration, we draw a clean sample $x_0$ from the data distribution $p(x_0)$, a uniformly sampled timestep $t \sim U[1, T]$, and Gaussian noise $\epsilon \sim \mathcal{N}(0, I)$. We forward noise the sample using equation 1 and then predict the injected noise using the denoiser $\epsilon_\theta$. Parameters are updated by minimizing the mean squared error with the MSE loss, following the standard training protocol, as proposed in Ho et al. (2020).

---

**Algorithm 2** Training of Unconditional Diffusion Model

---

1: **repeat**
2:     Sample data point $x_0 \sim p(x_0)$
3:     Sample timestep $t \sim \text{Uniform}(\{1, \ldots, T\})$
4:     Sample noise $\epsilon \sim \mathcal{N}(0, I)$
5:     Predict noise using backbone $\epsilon_\theta(\sqrt{\bar{\alpha}_t}x_0 + \sqrt{1 - \bar{\alpha}_t}\,\epsilon, t)$
6:     Update parameters with gradient descent on

$$\nabla_\theta \big\| \epsilon - \epsilon_\theta(x_t, t) \big\|_2^2$$

7: **until** convergence

---

### H.2 SAMPLING WITH HARPOON

At inference time, HARPOON augments the standard denoising procedure with manifold-aware conditional guidance (see Algorithm 1). We initialize the sample with Gaussian noise $x_T \sim \mathcal{N}(0, I)$. At denoising each step, we predict an estimate $\hat{x}_0$ of the clean sample. A task-specific conditional

loss $\mathcal{L}(\hat{x}_0, c)$ is then evaluated, and its gradient with respect to $x_t$ is computed. This gradient serves as an approximate tangential direction along the data manifold $\mathcal{M}_t$. Since the noisy manifolds at successive timesteps are approximately parallel, the gradient computed at timestep $t$ provides a good local approximation of the tangent space at $t-1$. HARPOON therefore reuses these gradients across steps to maintain manifold alignment. Concretely, after performing the standard denoising update to obtain a provisional sample $x'_{t-1}$, HARPOON applies an additional tangential correction:

$$x_{t-1} = x'_{t-1} - \eta \, \nabla_{x_t} \mathcal{L}(x_t, c),$$

where $\eta$ is a step size hyperparameter. This ensures that updates both (i) respect the manifold structure learned during training and (ii) steer the sample toward satisfying the imposed condition.

## I ADDITIONAL EXPERIMENTS

### I.1 IMPUTATION RESULTS UNDER MCAR AND MNAR MASKS

Beyond the main results with MAR masks (Sec. 4.2), we further evaluate Harpoon and baseline methods under MCAR and MNAR missingness in Tables 10,11,12, and 13. We notice that the overall trends remain consistent with the MAR setting. In terms of MSE and accuracy, DiffPuter and HARPOON generally perform the best, with HARPOON generally being the better among the two. While methods such as Remasker and Miracle are also strong baselines, we see that their bias towards numeric-type data limits their accuracy when imputing discrete features (see Tables 11,13).

Table 10: Imputation MSE for MCAR mask. Standard deviation in subscript.

| Ratio | Method | Adult | Bean | California | Default | Gesture | Letter | Magic | Shoppers |
|---|---|---|---|---|---|---|---|---|---|
| 0.25 | GAIN | $1.77_{0.04}$ | $1.27_{0.02}$ | $13.75_{0.10}$ | $5.67_{0.03}$ | $1.16_{0.02}$ | $1.03_{0.01}$ | $1.46_{0.02}$ | $1.82_{0.03}$ |
| | Miracle | $1.12_{0.04}$ | $\mathbf{0.08}_{0.00}$ | $\mathbf{0.38}_{0.02}$ | $0.83_{0.11}$ | $0.86_{0.05}$ | $\underline{0.45}_{0.03}$ | $\mathbf{0.51}_{0.03}$ | $1.27_{0.10}$ |
| | GReaT | $5.12_{4.38}$ | $2.73_{1.85}$ | $1.67_{0.04}$ | $1.75_{0.26}$ | $>10^6$ | $1.74_{0.01}$ | $1.60_{0.01}$ | $1.63_{0.07}$ |
| | Remasker | $\mathbf{0.94}_{0.05}$ | $0.26_{0.09}$ | $1.01_{0.12}$ | $\underline{0.51}_{0.03}$ | $0.67_{0.07}$ | $0.65_{0.07}$ | $1.70_{1.30}$ | $0.86_{0.11}$ |
| | DiffPuter | $1.05_{0.05}$ | $0.16_{0.01}$ | $0.49_{0.02}$ | $0.53_{0.03}$ | $\mathbf{0.34}_{0.01}$ | $\mathbf{0.39}_{0.01}$ | $\underline{0.67}_{0.02}$ | $\underline{0.80}_{0.05}$ |
| | Harpoon | $\underline{1.01}_{0.06}$ | $\underline{0.15}_{0.01}$ | $\underline{0.44}_{0.01}$ | $\mathbf{0.45}_{0.02}$ | $\underline{0.47}_{0.01}$ | $0.47_{0.01}$ | $0.79_{0.02}$ | $\mathbf{0.74}_{0.03}$ |
| 0.50 | GAIN | $3.30_{0.02}$ | $1.79_{0.01}$ | $17.30_{0.13}$ | $19.16_{0.07}$ | $3.06_{0.02}$ | $1.11_{0.01}$ | $1.82_{0.00}$ | $4.64_{0.03}$ |
| | Miracle | $4.00_{0.56}$ | $\mathbf{0.16}_{0.02}$ | $0.83_{0.05}$ | $1.37_{0.09}$ | $1.21_{0.02}$ | $0.89_{0.04}$ | $1.63_{0.25}$ | $2.28_{0.20}$ |
| | GReaT | $2.50_{1.38}$ | $4.09_{3.74}$ | $1.55_{0.04}$ | $2.30_{1.47}$ | $>10^5$ | $1.55_{0.02}$ | $1.40_{0.02}$ | $1.58_{0.34}$ |
| | Remasker | $\mathbf{1.06}_{0.02}$ | $1.02_{0.19}$ | $1.40_{0.02}$ | $0.86_{0.05}$ | $1.91_{0.44}$ | $1.12_{0.05}$ | $4.37_{0.02}$ | $1.23_{0.04}$ |
| | DiffPuter | $1.22_{0.03}$ | $0.27_{0.01}$ | $\underline{0.77}_{0.02}$ | $\underline{0.64}_{0.02}$ | $\mathbf{0.54}_{0.01}$ | $\mathbf{0.70}_{0.02}$ | $0.81_{0.00}$ | $\underline{0.93}_{0.03}$ |
| | Harpoon | $\underline{1.12}_{0.02}$ | $\underline{0.18}_{0.01}$ | $\mathbf{0.70}_{0.01}$ | $\mathbf{0.53}_{0.02}$ | $\underline{0.70}_{0.02}$ | $\underline{0.83}_{0.02}$ | $\underline{0.87}_{0.01}$ | $\mathbf{0.87}_{0.02}$ |
| 0.75 | GAIN | $6.63_{0.01}$ | $2.83_{0.01}$ | $21.66_{0.08}$ | $58.09_{0.22}$ | $9.59_{0.05}$ | $1.29_{0.00}$ | $2.36_{0.01}$ | $13.07_{0.05}$ |
| | Miracle | $1.47_{0.14}$ | $\underline{0.53}_{0.02}$ | $\mathbf{0.89}_{0.06}$ | $>10^4$ | $1.23_{0.04}$ | $\mathbf{0.86}_{0.01}$ | $\mathbf{0.96}_{0.05}$ | $\underline{1.13}_{0.05}$ |
| | GReaT | $1.79_{0.58}$ | $4.21_{1.73}$ | $1.50_{0.01}$ | $2.34_{2.01}$ | $>10^5$ | $1.50_{0.01}$ | $1.27_{0.01}$ | $1.33_{0.09}$ |
| | Remasker | $\mathbf{1.07}_{0.01}$ | $1.31_{0.00}$ | $1.40_{0.00}$ | $1.01_{0.01}$ | $6.66_{0.01}$ | $\underline{1.11}_{0.00}$ | $4.38_{0.02}$ | $1.13_{0.01}$ |
| | DiffPuter | $1.47_{0.02}$ | $0.63_{0.02}$ | $1.19_{0.01}$ | $\underline{0.90}_{0.01}$ | $\mathbf{0.82}_{0.02}$ | $1.18_{0.01}$ | $1.12_{0.01}$ | $1.20_{0.03}$ |
| | Harpoon | $\underline{1.28}_{0.01}$ | $\mathbf{0.39}_{0.00}$ | $\underline{1.11}_{0.01}$ | $\mathbf{0.72}_{0.01}$ | $\underline{0.97}_{0.02}$ | $1.27_{0.02}$ | $\underline{1.09}_{0.01}$ | $\mathbf{1.06}_{0.02}$ |

### I.2 ABLATION: EFFECT OF ENCODING

In Tables 14 and 15, we compare different encoding schemes on imputation performance, measured by MSE (for continuous features) and accuracy (for discrete features). Specifically, we consider *one-hot encoding* versus *integer encoding*, where the latter assigns a unique integer to each category of a feature. Our results show that one-hot encoding consistently outperforms integer encoding, both in terms of MSE and discrete imputation accuracy. We attribute this to the fact that integer encoding implicitly imposes an artificial ordering among categories, which may bias the model towards predicting categories with numerically closer values, even if they are semantically unrelated. In contrast, one-hot encoding avoids this issue by representing categories in an unordered manner.

### I.3 CHOICE OF PREPROCESSING

Table 16 shows the imputation performance under MAR missingness for three datasets when using either standard scaling (Z-normalization) or quantile standardization to preprocess continuous

Table 11: Imputation Accuracy for MCAR mask. Standard deviation in subscript.

| Ratio | Method | Adult | Default | Shoppers |
|---|---|---|---|---|
| 0.25 | GAIN | $20.77_{0.21}$ | $47.77_{0.43}$ | $45.43_{0.43}$ |
| | Miracle | $43.48_{0.89}$ | $65.23_{2.07}$ | $40.22_{0.47}$ |
| | GReaT | $15.90_{0.77}$ | $41.74_{0.27}$ | $40.51_{0.46}$ |
| | Remasker | $41.36_{0.57}$ | $67.03_{0.71}$ | $42.19_{0.34}$ |
| | DiffPuter | $\underline{66.47}_{0.23}$ | $\underline{69.91}_{0.26}$ | $\mathbf{59.11}_{0.37}$ |
| | Harpoon | $\mathbf{66.54}_{0.19}$ | $\mathbf{72.22}_{0.30}$ | $\underline{54.67}_{0.55}$ |
| 0.50 | GAIN | $19.94_{0.15}$ | $41.14_{0.12}$ | $32.45_{0.20}$ |
| | Miracle | $30.24_{1.19}$ | $56.03_{1.35}$ | $32.50_{1.26}$ |
| | GReaT | $31.08_{0.87}$ | $49.99_{0.44}$ | $39.79_{0.38}$ |
| | Remasker | $33.28_{3.21}$ | $51.23_{2.31}$ | $43.76_{1.00}$ |
| | DiffPuter | $\underline{60.59}_{0.21}$ | $\underline{61.71}_{0.46}$ | $\mathbf{55.90}_{0.23}$ |
| | Harpoon | $\mathbf{61.95}_{0.21}$ | $\mathbf{68.25}_{0.33}$ | $\underline{52.26}_{0.31}$ |
| 0.75 | GAIN | $21.22_{0.06}$ | $19.99_{0.18}$ | $19.27_{0.13}$ |
| | Miracle | $33.68_{0.74}$ | $49.02_{10.05}$ | $38.02_{0.56}$ |
| | GReaT | $42.62_{0.25}$ | $\underline{54.91}_{0.33}$ | $38.17_{0.33}$ |
| | Remasker | $29.17_{0.04}$ | $44.45_{0.14}$ | $46.59_{0.09}$ |
| | DiffPuter | $\underline{53.82}_{0.17}$ | $49.64_{0.34}$ | $\mathbf{51.77}_{0.40}$ |
| | Harpoon | $\mathbf{55.32}_{0.06}$ | $\mathbf{60.55}_{0.27}$ | $\underline{49.65}_{0.36}$ |

Table 12: Imputation MSE for MNAR mask. Standard deviation in subscript.

| Ratio | Method | adult | bean | california | default | gesture | letter | magic | shoppers |
|---|---|---|---|---|---|---|---|---|---|
| 0.25 | GAIN | $1.86_{0.14}$ | $1.35_{0.08}$ | $13.92_{0.48}$ | $5.79_{0.20}$ | $1.30_{0.06}$ | $1.03_{0.03}$ | $1.48_{0.10}$ | $1.77_{0.13}$ |
| | Miracle | $1.33_{0.16}$ | $0.27_{0.15}$ | $0.55_{0.20}$ | $0.95_{0.15}$ | $1.04_{0.07}$ | $0.54_{0.05}$ | $\mathbf{0.65}_{0.13}$ | $1.60_{0.30}$ |
| | GReaT | $2.83_{1.58}$ | $1.96_{0.15}$ | $1.76_{0.11}$ | $1.99_{0.20}$ | $>10^5$ | $1.75_{0.03}$ | $1.67_{0.08}$ | $5.26_{7.18}$ |
| | Remasker | $\mathbf{0.96}_{0.09}$ | $0.27_{0.09}$ | $1.18_{0.23}$ | $\underline{0.61}_{0.11}$ | $0.77_{0.07}$ | $0.69_{0.11}$ | $1.09_{0.09}$ | $0.90_{0.13}$ |
| | DiffPuter | $1.05_{0.07}$ | $\underline{0.19}_{0.06}$ | $\underline{0.51}_{0.07}$ | $0.64_{0.09}$ | $\mathbf{0.46}_{0.07}$ | $\mathbf{0.39}_{0.02}$ | $\underline{0.73}_{0.10}$ | $\underline{0.84}_{0.07}$ |
| | Harpoon | $\underline{1.02}_{0.07}$ | $\mathbf{0.15}_{0.03}$ | $0.46_{0.04}$ | $0.51_{0.10}$ | $\underline{0.60}_{0.11}$ | $\underline{0.49}_{0.02}$ | $0.88_{0.15}$ | $\mathbf{0.78}_{0.06}$ |
| 0.50 | GAIN | $3.32_{0.08}$ | $1.76_{0.05}$ | $17.40_{0.34}$ | $18.82_{0.14}$ | $3.08_{0.05}$ | $1.09_{0.04}$ | $1.80_{0.07}$ | $4.60_{0.18}$ |
| | Miracle | $3.96_{0.52}$ | $0.39_{0.13}$ | $0.92_{0.17}$ | $1.30_{0.17}$ | $1.23_{0.06}$ | $0.96_{0.06}$ | $1.47_{0.15}$ | $2.37_{0.23}$ |
| | GReaT | $2.93_{1.77}$ | $2.52_{0.75}$ | $2.00_{0.35}$ | $1.73_{0.39}$ | $>10^5$ | $1.55_{0.02}$ | $1.40_{0.05}$ | $1.41_{0.06}$ |
| | Remasker | $\mathbf{1.04}_{0.07}$ | $1.12_{0.22}$ | $1.42_{0.09}$ | $0.79_{0.08}$ | $2.14_{0.30}$ | $1.14_{0.04}$ | $4.35_{0.15}$ | $1.18_{0.03}$ |
| | DiffPuter | $1.22_{0.07}$ | $\underline{0.30}_{0.08}$ | $\underline{0.76}_{0.06}$ | $\underline{0.66}_{0.07}$ | $\mathbf{0.57}_{0.06}$ | $\mathbf{0.69}_{0.02}$ | $\mathbf{0.81}_{0.06}$ | $\underline{0.91}_{0.04}$ |
| | Harpoon | $\underline{1.12}_{0.07}$ | $\mathbf{0.19}_{0.03}$ | $0.71_{0.05}$ | $0.54_{0.07}$ | $\underline{0.74}_{0.08}$ | $0.82_{0.02}$ | $\underline{0.85}_{0.09}$ | $0.83_{0.01}$ |
| 0.75 | GAIN | $6.83_{0.41}$ | $2.85_{0.07}$ | $21.70_{0.24}$ | $58.61_{0.70}$ | $9.81_{0.07}$ | $1.28_{0.05}$ | $2.33_{0.03}$ | $13.42_{0.56}$ |
| | Miracle | $1.71_{0.23}$ | $0.80_{0.05}$ | $\mathbf{0.95}_{0.06}$ | $0.94_{0.08}$ | $1.25_{0.05}$ | $\mathbf{1.00}_{0.03}$ | $\underline{1.05}_{0.06}$ | $1.17_{0.06}$ |
| | GReaT | $1.48_{0.06}$ | $7.91_{4.18}$ | $1.62_{0.28}$ | $1.89_{0.31}$ | $>10^5$ | $1.50_{0.02}$ | $1.25_{0.06}$ | $1.30_{0.05}$ |
| | Remasker | $\mathbf{1.05}_{0.03}$ | $1.29_{0.06}$ | $1.39_{0.02}$ | $0.95_{0.06}$ | $6.63_{0.04}$ | $\underline{1.10}_{0.01}$ | $4.33_{0.06}$ | $\underline{1.13}_{0.02}$ |
| | DiffPuter | $1.47_{0.06}$ | $\underline{0.61}_{0.09}$ | $1.18_{0.02}$ | $\underline{0.85}_{0.08}$ | $\mathbf{0.79}_{0.05}$ | $1.17_{0.01}$ | $1.09_{0.06}$ | $1.22_{0.04}$ |
| | Harpoon | $\underline{1.29}_{0.05}$ | $\mathbf{0.38}_{0.05}$ | $\underline{1.10}_{0.03}$ | $\mathbf{0.67}_{0.05}$ | $\underline{0.91}_{0.05}$ | $1.25_{0.02}$ | $\mathbf{1.04}_{0.09}$ | $\mathbf{1.07}_{0.03}$ |

features[10]. Across all datasets and missingness ratios, we observe that standard scaling provides consistently lower MSE on continuous features compared to the quantile counterpart, sometimes by more than an order of magnitude. In contrast, categorical imputation accuracy remains largely unchanged, since categorical variables are not affected by feature scaling.

The large errors with quantile scaling are probably because of highly non-linear mappings that reshape features to match a uniform (or normal) distribution, which can produce large jumps in the data space even for small differences in the transformed space, especially for sparse or heavy-tailed regions of the distribution. Hence, we see that for Adult, where continuous features are relatively well-behaved, quantile scaling only slightly worsens performance. However, in the Default and Shoppers datasets, both of which contain skewed or heavy-tailed numerical variables, the quantile mapping results in large MSE errors.

---

[10]We use the default preprocessor from `https://scikit-learn.org/stable/modules/generated/sklearn.preprocessing.QuantileTransformer.html`

Table 13: Imputation Accuracy for MNAR mask. Standard deviation in subscript.

| Ratio | Method | adult | default | shoppers |
|---|---|---|---|---|
| 0.25 | GAIN | $20.62_{0.29}$ | $49.72_{2.10}$ | $44.56_{0.94}$ |
| | Miracle | $43.11_{1.76}$ | $63.90_{4.92}$ | $39.79_{0.71}$ |
| | GReaT | $16.49_{0.96}$ | $42.28_{0.62}$ | $39.43_{0.80}$ |
| | Remasker | $40.11_{2.94}$ | $66.65_{1.47}$ | $42.01_{1.00}$ |
| | DiffPuter | $\mathbf{67.02}_{1.31}$ | $\underline{70.50}_{0.66}$ | $\mathbf{58.85}_{0.80}$ |
| | Harpoon | $\underline{66.86}_{1.29}$ | $\mathbf{72.58}_{0.64}$ | $\underline{53.94}_{0.71}$ |
| 0.50 | GAIN | $19.89_{0.12}$ | $42.03_{1.43}$ | $32.06_{0.56}$ |
| | Miracle | $31.11_{1.32}$ | $54.32_{1.64}$ | $33.15_{1.47}$ |
| | GReaT | $31.66_{0.90}$ | $50.26_{0.47}$ | $38.94_{0.36}$ |
| | Remasker | $35.41_{0.16}$ | $55.23_{1.81}$ | $44.29_{1.65}$ |
| | DiffPuter | $\underline{60.57}_{0.84}$ | $\underline{62.10}_{0.65}$ | $\mathbf{55.61}_{0.90}$ |
| | Harpoon | $\mathbf{61.92}_{0.85}$ | $\mathbf{68.48}_{0.45}$ | $\underline{51.85}_{0.89}$ |
| 0.75 | GAIN | $20.94_{0.44}$ | $20.22_{1.01}$ | $19.13_{0.56}$ |
| | Miracle | $31.50_{2.89}$ | $53.44_{1.90}$ | $39.18_{1.06}$ |
| | GReaT | $42.76_{0.83}$ | $\underline{54.85}_{0.36}$ | $38.35_{0.38}$ |
| | Remasker | $28.95_{0.30}$ | $43.89_{1.02}$ | $46.33_{0.16}$ |
| | DiffPuter | $\underline{53.33}_{0.63}$ | $49.29_{0.44}$ | $\mathbf{51.56}_{0.61}$ |
| | Harpoon | $\mathbf{54.93}_{0.79}$ | $\mathbf{60.40}_{0.64}$ | $\underline{49.53}_{0.43}$ |

Table 14: Imputation MSE (MAR mask) under different encodings for HARPOON.

| Ratio | Method | Adult | Default | Shoppers |
|---|---|---|---|---|
| 0.25 | One-Hot | $\mathbf{0.99}_{0.08}$ | $\mathbf{0.51}_{0.06}$ | $\mathbf{0.73}_{0.06}$ |
| | Integer | $1.61_{0.10}$ | $0.67_{0.10}$ | $1.05_{0.14}$ |
| 0.50 | One-Hot | $\mathbf{1.08}_{0.06}$ | $\mathbf{0.53}_{0.06}$ | $\mathbf{0.81}_{0.06}$ |
| | Integer | $1.58_{0.08}$ | $0.68_{0.08}$ | $1.09_{0.12}$ |
| 0.75 | One-Hot | $\mathbf{1.26}_{0.07}$ | $\mathbf{0.72}_{0.06}$ | $\mathbf{1.10}_{0.05}$ |
| | Integer | $1.54_{0.05}$ | $0.82_{0.10}$ | $1.26_{0.04}$ |

## I.4 RUNTIME COSTS

Table 17 shows the runtime costs of methods on all datasets. The observations are consistent with what was reported in the main text. We see that HARPOON's still stays within 2x the runtime cost of DiffPuter due to the doubling of function calls to the denoiser backbone for gradient propagation.

## I.5 RESULTS WITH ADDITIONAL METRICS

Table 18 reports the KS, detectability, and identifiability scores of synthetic datasets generated under different constraints: Range, Category, Conjunctions, and Disjunctions. Overall, KS scores remain consistently high across all methods and constraints, indicating that the synthetic data effectively preserves the marginal distributions of the original datasets.

Detectability, which measures how easily a classifier can distinguish synthetic from real data, varies with the type of constraint. Under stricter constraints, such as conjunctions and categorical, detectability is generally lower, indicating that samples are more indistinguishable from the real data. This is expected, as enforcing categorical or conjunctive constraints restricts the model to a subset of discrete labels or consistent combinations, producing samples that more closely resemble real data. In contrast, Range constraints allow more flexibility, since values can vary continuously within plausible bounds, which can increase detectability. For HARPOON in particular, detectability noticeably decreases under conjunctive and categorical constraints, suggesting that the model effectively guides samples toward regions satisfying both objectives.

Identifiability, which quantifies the risk of exposing real records, remains generally low across all methods. However, we see an interesting trade-off with the detectability. The more a sample resem-

Table 15: Imputation Accuracy (MAR mask) under different encodings for HARPOON.

| Ratio | Method | Adult | Default | Shoppers |
|-------|--------|-------|---------|----------|
| 0.25 | One-Hot | **69.44**$_{3.11}$ | **73.49**$_{2.53}$ | **51.17**$_{5.20}$ |
| | Integer | 65.04$_{3.22}$ | 73.02$_{2.48}$ | 45.12$_{4.48}$ |
| 0.50 | One-Hot | **64.80**$_{3.29}$ | **69.55**$_{1.91}$ | **48.39**$_{5.00}$ |
| | Integer | 59.82$_{2.74}$ | 68.86$_{1.64}$ | 43.16$_{4.80}$ |
| 0.75 | One-Hot | **58.81**$_{3.85}$ | 61.48$_{3.83}$ | **48.37**$_{4.39}$ |
| | Integer | 55.21$_{2.54}$ | **62.76**$_{2.73}$ | 45.45$_{3.50}$ |

Table 16: Imputation MSE (continuous features) and Accuracy (categorical features) under MAR mask and various preprocessing scalers.

| Ratio | Method | Adult | | Default | | Shoppers | |
|-------|--------|---------|---------|---------|---------|---------|---------|
| | | Avg. MSE | Avg. Acc | Avg. MSE | Avg. Acc | Avg. MSE | Avg. Acc |
| 0.25 | Standard | 0.99$_{0.08}$ | 69.44$_{3.11}$ | 0.51$_{0.06}$ | 73.49$_{2.53}$ | 0.73$_{0.06}$ | 51.17$_{5.20}$ |
| | Quantile | 1.03$_{0.10}$ | 66.60$_{3.34}$ | 12.19$_{2.52}$ | 74.88$_{2.50}$ | 5.79$_{1.80}$ | 50.86$_{5.40}$ |
| 0.50 | Standard | 1.08$_{0.06}$ | 64.80$_{3.29}$ | 0.53$_{0.06}$ | 69.55$_{1.91}$ | 0.81$_{0.06}$ | 48.39$_{5.00}$ |
| | Quantile | 1.14$_{0.11}$ | 63.25$_{3.36}$ | 13.69$_{3.47}$ | 70.78$_{2.08}$ | 5.56$_{1.55}$ | 48.30$_{5.66}$ |
| 0.75 | Standard | 1.26$_{0.07}$ | 58.81$_{3.85}$ | 0.72$_{0.06}$ | 61.48$_{3.83}$ | 1.10$_{0.05}$ | 48.37$_{4.39}$ |
| | Quantile | 1.43$_{0.16}$ | 57.81$_{4.24}$ | 20.45$_{9.37}$ | 61.92$_{4.16}$ | 4.78$_{0.99}$ | 48.60$_{4.78}$ |

bles real data (lower detectability), the higher the potential risk of revealing private records (higher identifiability). Range constraints enforce numerical consistency, resulting in low identifiability but moderate detectability due to skewed distributions of generated outputs. On the other hand, category constraints preserve categorical labels, lowering detectability while increasing identifiability. Combining both constraints in the AND setting increases the identifiability of HARPOON, but lowers the detectability. The OR setting provides a balance, producing realistic data with moderate privacy protection.

Table 17: Runtimes (seconds) of methods across datasets (MAR mask 0.25 missing ratio).

| Method | Adult | Shoppers | Default | Gesture | Magic | Bean | California | Letter |
|--------|-------|----------|---------|---------|-------|------|------------|--------|
| GAIN | 0.14$_{0.26}$ | 0.14$_{0.26}$ | 0.15$_{0.28}$ | 0.14$_{0.27}$ | 0.14$_{0.27}$ | 0.14$_{0.27}$ | 0.15$_{0.29}$ | 0.14$_{0.27}$ |
| Miracle | 33.03$_{0.99}$ | 18.27$_{0.30}$ | 44.54$_{0.29}$ | 23.09$_{0.13}$ | 19.99$_{0.11}$ | 19.86$_{0.13}$ | 21.31$_{0.06}$ | 26.39$_{0.29}$ |
| GReaT | 65.47$_{2.57}$ | 51.13$_{2.29}$ | 202.71$_{7.44}$ | 131.72$_{1.96}$ | 19.84$_{0.82}$ | 75.29$_{4.32}$ | 25.05$_{1.30}$ | 103.68$_{0.56}$ |
| Remasker | 2.68$_{5.24}$ | 0.28$_{0.51}$ | 0.33$_{0.53}$ | 0.30$_{0.54}$ | 0.30$_{0.53}$ | 0.29$_{0.53}$ | 0.29$_{0.52}$ | 0.30$_{0.52}$ |
| DiffPuter | 2.83$_{0.27}$ | 1.19$_{0.25}$ | 2.62$_{0.26}$ | 0.98$_{0.27}$ | 1.68$_{0.26}$ | 1.32$_{0.26}$ | 1.86$_{0.26}$ | 1.79$_{0.25}$ |
| Harpoon | 4.43$_{0.34}$ | 1.89$_{0.36}$ | 4.05$_{0.34}$ | 1.54$_{0.35}$ | 2.65$_{0.34}$ | 2.08$_{0.34}$ | 2.88$_{0.34}$ | 2.80$_{0.33}$ |

Table 18: KS, detectability and identifiability scores.

| Constr. | Method | Adult | | | Default | | | Shoppers | | |
|---|---|---|---|---|---|---|---|---|---|---|
| | | KS (↑) % | Det. (↓) | Iden. (↓) | KS (↑) % | Det. (↓) | Iden. (↓) | KS (↑) % | Det. (↓) | Iden. (↓) |
| *Range* | DiffPuter | $0.94_{0.00}$ | $0.94_{0.00}$ | $0.17_{0.01}$ | $0.93_{0.00}$ | $0.94_{0.01}$ | $0.15_{0.01}$ | $0.91_{0.00}$ | $0.94_{0.00}$ | $0.29_{0.02}$ |
| | GReaT | $0.96_{0.00}$ | $0.95_{0.00}$ | $0.20_{0.01}$ | $0.92_{0.00}$ | $0.99_{0.00}$ | $0.08_{0.01}$ | $0.91_{0.00}$ | $0.99_{0.00}$ | $0.07_{0.01}$ |
| | Harpoon | $0.95_{0.00}$ | $0.90_{0.00}$ | $0.26_{0.00}$ | $0.94_{0.00}$ | $0.89_{0.00}$ | $0.31_{0.01}$ | $0.92_{0.00}$ | $0.87_{0.00}$ | $0.44_{0.01}$ |
| *Category* | DiffPuter | $0.95_{0.00}$ | $0.85_{0.01}$ | $0.41_{0.01}$ | $0.94_{0.00}$ | $0.75_{0.01}$ | $0.44_{0.01}$ | $0.89_{0.00}$ | $0.89_{0.00}$ | $0.41_{0.02}$ |
| | GReaT | $0.97_{0.00}$ | $0.75_{0.01}$ | $0.44_{0.03}$ | $0.93_{0.00}$ | $0.79_{0.01}$ | $0.29_{0.01}$ | $0.91_{0.00}$ | $0.91_{0.00}$ | $0.28_{0.02}$ |
| | Harpoon | $0.97_{0.00}$ | $0.73_{0.01}$ | $0.46_{0.02}$ | $0.95_{0.00}$ | $0.81_{0.01}$ | $0.40_{0.01}$ | $0.90_{0.00}$ | $0.86_{0.01}$ | $0.47_{0.01}$ |
| *Range* AND *Category* | DiffPuter | $0.94_{0.00}$ | $0.91_{0.01}$ | $0.30_{0.05}$ | $0.93_{0.01}$ | $0.95_{0.01}$ | $0.15_{0.01}$ | $0.88_{0.00}$ | $0.94_{0.01}$ | $0.27_{0.02}$ |
| | GReaT | $0.96_{0.00}$ | $0.92_{0.01}$ | $0.31_{0.04}$ | $0.92_{0.00}$ | $0.98_{0.01}$ | $0.07_{0.02}$ | $0.92_{0.00}$ | $0.99_{0.00}$ | $0.07_{0.01}$ |
| | Harpoon | $0.96_{0.00}$ | $0.64_{0.02}$ | $0.47_{0.06}$ | $0.94_{0.00}$ | $0.82_{0.00}$ | $0.38_{0.04}$ | $0.89_{0.00}$ | $0.82_{0.01}$ | $0.53_{0.01}$ |
| *Range* OR *Category* | DiffPuter | $0.94_{0.00}$ | $0.92_{0.00}$ | $0.22_{0.00}$ | $0.94_{0.00}$ | $0.80_{0.00}$ | $0.37_{0.00}$ | $0.90_{0.00}$ | $0.93_{0.00}$ | $0.30_{0.01}$ |
| | GReaT | $0.97_{0.00}$ | $0.90_{0.00}$ | $0.24_{0.00}$ | $0.93_{0.00}$ | $0.84_{0.01}$ | $0.24_{0.01}$ | $0.92_{0.00}$ | $0.98_{0.00}$ | $0.14_{0.01}$ |
| | Harpoon | $0.95_{0.00}$ | $0.92_{0.00}$ | $0.23_{0.01}$ | $0.92_{0.00}$ | $0.94_{0.00}$ | $0.20_{0.00}$ | $0.91_{0.00}$ | $0.90_{0.00}$ | $0.37_{0.01}$ |

