# OpenReview forum: "Harpoon: Generalised Manifold Guidance for Conditional Tabular Diffusion"
_ICLR.cc/2026/Conference — ICLR 2026 Poster_

### Official Review · Reviewer_1jxC · 2025-10-29

**Soundness:** 3
**Presentation:** 2
**Contribution:** 2
**Rating:** 4
**Confidence:** 3

**Summary:**

The paper generalizes previous ideas from manifold theory to diffusion models for tabular data. This allows the usage of differentiable losses at inference time to guide samples along the data manifold at a given diffusion time $t$. Depending on the objective, this produces samples that satisfy constraints based on partially observed data and specified inequality constraints without the need for re-training the diffusion model.

**Strengths:**

- The authors generalize previous results, extending the usefulness of manifold-based insights to tabular data. In particular, they remove the necessity of squared losses, which extends the effective conditioning capabilities.
- The ability to condition on certain features or constraints at inference time without the need for training the model on that specific conditional generation task is very valuable, in particular for tabular data.
- The illustrations mostly paint an intuitive picture of the underlying mechanisms.
- I like the effort of making the model more comparable to other diffusion-based approaches by adjusting the backend architecture.
- The presented results show competitive or better performance in practice. Great improvements can be seen in scenarios that impose inequality constraints.

**Weaknesses:**

- The methodological background on diffusion is basically non-existent and the overview of the tabular diffusion models is severely underdeveloped. Since TabDDPM, many models have been proposed that 1) often considerably outperform TabDDPM and 2) do not rely on multinomial diffusion. In fact, models that treat categorical data in some continuous space exist (e.g., TabSyn [1] or CDTD [2]). It is unclear how the results extend to such models.
- The missingness rates of 0.5 and 0.75 in the main results seem very extreme. The cited DiffPuter paper uses 30%. In reality, it is questionable why we would want to impute data when 75% of the data is actually missing. For more moderate missing rates, the performance advantage of the proposed method is not clear but it remains mostly competitive.
-  Only using a single metric to evaluate sample quality of tabular data (in the case of inequality constraints) is not enough. Metrics like the detection score (see, e.g., metrics used in [1] and [2]) which evaluate the joint distribution more holistically would be interesting to see and give more insight into how the manifold guidance impacts samples.
- The examined constraints for the conditional generation tasks are rather simple and focus on a maximum of two features (one categorical, one continuous), so the proportion that training samples are valid is still quite high. It is unclear how the framework performs under more constraints, which make the guidance more difficult. An ablation study should investigate what happens if the number of constraints is increased, such that the number of valid training samples shrinks. In the extreme case, is the method able to recover a single valid observation?
- The paper makes the claim in line 112 that extending the results from Chung et al. to tabular diffusion is not trivial. One reason is that tabular data contains discrete features. This, however, is then solved by the rather trivial solution of simply one-hot encoding discrete features and treating them as continuous. Besides this encoding, there is no tabular-data-specific modeling in the paper.
- Figure 2a) does not actually illustrate the behavior of a "spotlight" that becomes sharper near $\mathcal{M}_0$.  In the Figure the position of $x_t$ relative to $\mathcal{M}_0$ never changes. When the spotlight becomes sharper, I would expect an $x_s$ closer to $\mathcal{M}_0$ where $s < t$. Note also that the orthogonal projection of $x_t$ and $x_s$ need not be the same.
- Making the assumption in line 253 that $Q_t(x_t)$ is approximately orthogonal for all $t$ based on only a single empirical observation seems not well-founded. Could this assumption not be weakened by making $\eta$ time-dependent, such that the guidance term has more impact when it is also more likely to hold, i.e., as $t\rightarrow 0$?
- It is not clear what tabular diffusion framework is assumed. It appears to be very similar to to StaSy model [3], which also one-hot encodes categorical data. The authors should be more specific about the framework they are using. If it can be used on any or most tabular diffusion frameworks (maybe under certain conditions) this should be stated as a strength of the framework. Coming up with an entirely new generative model does not highlight the strength of the potentially more general approach.
- Getting the tangential gradient requires backpropagating through $\epsilon_\theta$. It should be clarified how costly the method is, in particular at inference time and relative to the other baselines.
- Based on the inference time losses and the results, it should be highlighted by the authors that it is not guaranteed that samples satisfy the specified constraints. The approach is more similar to penalization than imposing hard constraints.
- The authors do not discuss how their approach differs from classifier-guidance when using similar losses.

---


[1] Zhang, et al. (2024) Mixed-Type Tabular Data Synthesis with Score-based Diffusion in Latent Space. ICLR.

[2] Mueller, et al. (2025) Continuous Diffusion for Mixed-Type Tabular Data. ICLR.

[3] Kim, et al. (2023) STaSy: Score-based Tabular data Synthesis. ICLR.

**Questions:**

In addition to the suggestions and questions stated in Weaknesses, I kindly ask the following questions:

- Could this manifold guidance lead to a skewed (conditional) data distribution?
- Your guidance mechanism leads to *soft* constraints but not cannot impose *hard* constraints. This should be made clear in the text. Does that lead to issues in the imputation task? Since the non-missing values are not strictly fixed but still updated during sampling, could it lead to the final samples actually deviating from the partially observed information?
- Does the guidance also work for latent diffusion models (TabSyn, CDTD, etc.) or is a data-space model necessarily needed?
Why is it a problem if models push samples orthogonally towards $\mathcal{M}_0$ (line 104/105)? Even if it skips a shell, in the next update step, the model, which is conditioned on $t$, should have no problem continuing the path if trained properly. Is this an artifact from a model that is not capable / complex enough?
- How does your guidance generalize to high dimensions, in particular when increasing the number of features or categories in a dataset? One-hot encoding is known to be very inefficient and will blow up the dimensions when applied to a categorical feature with 100s or 1000s of categories.
- Can relative or logical constraints be accommodated? For instance, consider a data table of order and delivery dates. Naturally, we would expect that order date < delivery date.
- Figure 2 b) and c) give a good intuition of moving a sample along a shell. Can there be situations in practice where this fails? For example, if at a particular level the manifolds are no longer connected, e.g., $\mathcal{M}_t$ is not a single piece but two separate ones?
- Are you using the QuantileTransformer to transform the continuous features for the TabDDPM model? From own experience this appears to be a crucial step for performance and is ubiquitous in tabular diffusion models.
- In line 420: "not just the constrained ones [...]", do you actually mean the *un*constrained features?
- Figure 3 is not well-designed. It does not does not include the red or green lines due to unfortunate overlaps. It would yield a more consistent argument if the x-axis would indicate $t \in [0,1]$ and not the denoising step.

---

> ### Author Response · Authors · 2025-11-16
> **Response by Authors (Weaknesses W1 - W4)**
>
> ### Weaknesses
> > W1: Background on (tabular) diffusion is non-existent. It is unclear how the results extend to SOTA latent models such as TabSyn and CDTD.
>
> [AW1] We will add more background details on diffusion and tabular diffusion models. Harpoon's guidance is currently limited to data-space models and we leave the extension to latent-space models as future work. Conditional generation with latent models is difficult because it requires mapping constraints from the data space to latent space, which should then be injected to guide sampling within the latent space. But this mapping is difficult if the constraints are only partially observed values or inequalities on features.
>
> > W2: The missingness rates of 0.5 and 0.75 seem very extreme. Unclear on the moderate missing rates.
>
> [AW2] The missingness rates of 0.5 and 0.75 were chosen to stress-test Harpoon under challenging scenarios, highlighting its robustness where other methods begin to struggle. In **Table F**, we see that Harpoon is among the top two methods (MSE and Accuracy) even for smaller missing rates (15%). Full results are here [ https://anonymous.4open.science/r/ManifoldTabularImputation-44E4/experiments/imputation_0.15_MAR.csv ].
>
> | Dataset | Method | Avg MSE | Avg Acc |
> | --- | ---  | --- | --- |
> |  | Harpoon | 0.957 | 71.131 |
> |  | GReaT | 5.745 | 14.266 |
> | Adult | DiffPuter | 0.991 | 71.720 |
> |  | Remasker | 0.868 | 50.996 |
> |  | Gain | 1.574 | 22.734 |
> |  | Miracle | 0.978 | 51.740 |
> |
> |  | Harpoon | 0.592 | 74.659 |
> |  | GReaT | 1.999 | 42.010 |
> | Default | DiffPuter | 0.722 | 73.455 |
> |  | Remasker | 0.658 | 71.603 |
> |  | Gain | 4.763 | 43.774 |
> |  | Miracle | 0.939 | 71.436 |
> |
> |  | Harpoon | 0.736 | 51.959 |
> |  | GReaT | 8.977 | 37.802 |
> | Shoppers | DiffPuter | 0.770 | 57.655 |
> |  | Remasker | 0.675 | 40.735 |
> |  | Gain | 1.363 | 43.157 |
> |  | Miracle | 1.083 | 36.765 |
>
> **Table F. Results on MAR 0.15 mask, averaged over 5 trials.**
>
> > W3:  Metrics which evaluate the joint distribution more holistically would be interesting to see.
>
> [AW3] **Table G** shows the results for range constraints with additional metrics using Synthcity [ https://github.com/vanderschaarlab/synthcity ]. Full results are here [ https://anonymous.4open.science/r/ManifoldTabularImputation-44E4/experiments/general_constraints_updated_utility.csv ]. High KS scores indicate that Harpoon captures the marginal distributions of real data well. Low identifiability indicates that the model does not memorise real samples, yielding better privacy. While high detectability indicates that the models have a selective bias, we still see that downstream utility remains strong (XGBoost classifier).
>
> |Dataset|Method|Avg xgb utility|Avg Alpha-P|Avg ViolationAcc|Avg ks|Avg detect|Avg identifiability|
> |-|-|-|-|-|-|-|-|
> ||GReaT| 0.947|0.922|83.577|0.962|0.953|0.196|
> |adult|Harpoon| 0.945|0.918|3.065|0.951|0.905|0.256|
> ||DiffPuter| 0.941|0.829|73.861|0.937|0.944|0.169|
> ||GReaT| 0.696|0.670|95.366|0.915|0.987|0.077|
> |default|Harpoon| 0.855|0.787|7.782|0.938|0.886|0.307|
> ||DiffPuter| 0.778|0.649|88.525|0.935|0.942|0.155|
> ||GReaT| 0.348|0.877|96.525|0.911|0.994|0.071|
> |shoppers|Harpoon| 0.886|0.604|20.545|0.922|0.874|0.436|
> ||DiffPuter| 0.892|0.684|76.687|0.907|0.937|0.292|
>
> **Table G. Additional metrics on range constraints.**
>
> > W4:  What happens if the number of constraints is increased. In the extreme case, can the method recover a single valid observation?
>
> [AW4] We specifically chose our constraints to minimise the number of valid training samples. In **Table 7 Appendix E**, we show that the constraints correspond to the tail-end of feature distributions. This results in the valid proportions of training samples as low as **0.87%** (Adult dataset, combined constraints), illustrating that Harpoon is effective under rare scenarios. To further stress-test under extreme cases, **Table H** provides all features as constraints and measures the MSE (continuous features) and accuracy (discrete features) w.r.t the regenerated data. We see that MSE scores are small and accuracy is almost 100%, indicating strong recovery. Full results are here [ https://anonymous.4open.science/r/ManifoldTabularImputation-44E4/experiments/extremeconstraint.csv ].
>
> |Dataset|Avg. MSE|Avg. Accuracy|
> |-|-|-|
> |Adult|0.115|99.930|
> |Bean|0.011|N.A|
> |California|0.012|N.A|
> |Default|0.024|99.999|
> |Gesture|0.015|N.A|
> |Letter|0.012|N.A|
> |Magic|0.012|N.A|
> |Shoppers|0.047|100.0|
>
> **Table H. Accuracy and MSE of recovered data when given all features as conditions.**

---

> ### Author Response · Authors · 2025-11-16
> **Response by Authors (Weaknesses W5-W11)**
>
> > W5: There is no tabular-data-specific modeling in the paper apart from one-hot encoding.
>
> [AW5] The tabular-specific aspects of Harpoon are introduced at **inference time**, which is our main contribution. **Table 4 and Remark 2** in the paper show that applying a **sparsity-inducing loss** during inference helps the model better handle one-hot encoded features, even improving over the training objective (MSE). The training of Harpoon is intentionally simple and tabular-data-agnostic and uses the common MSE objective. This design broadens applicability across a wide variety of architectures, including time series and image diffusion which use the MSE loss.
>
> > W6: Figure 2a) does not actually show increasing sharpness as t decreases.
>
> [AW6] We have added a new figure here [ https://anonymous.4open.science/r/ManifoldTabularImputation-44E4/new_figures/spotlight.pdf ] and would like to know if this conveys what the reviewer had in mind?
>
> > W7: Can the orthogonality assumption for all t be weakened by making eta time-dependent, i.e., more guidance as t approaches 0?
>
> [AW7] **Table I** presents the ablation results using a linearly increasing guidance strength as $t$ decreases. We find the the performance does not change by much, indicating that the approximation does hold quite well. Full results are here [ https://anonymous.4open.science/r/ManifoldTabularImputation-44E4/experiments/imputation_linear_schedule_ablation.csv ]. We have also added orthogonality plots for all datasets, and see that the pattern holds [ https://anonymous.4open.science/r/ManifoldTabularImputation-44E4/experiments/tubular_region_plots/gradient_angles_default.pdf ].
>
> | Dataset | Method | Avg MSE | Avg Acc |
> | --- | ---  | --- | --- |
> |  | Linear | 0.989 | 68.906 |
> | adult | Fixed | 0.995 | 69.438 |
> |
> |  | Linear | 0.509 | 73.209 |
> |default|Fixed|0.512|73.489|
> |
> |  |Linear|0.679|50.697|
> |shoppers|Fixed|0.732|51.170|
>
> **Table I. Ablation on Linear vs fixed guidance strength using MAR mask 0.25 ratio.**
>
> > W8: It is not clear what tabular diffusion framework is assumed, and if it can be used on any or most tabular diffusion frameworks.
>
> [AW8] Harpoon uses the backbone architecture proposed by DiffPuter, which also employs one-hot encoding for categorical features and trains using an MSE reconstruction loss. This was done to broaden applicability since MSE loss is also popular for image and time series diffusion. One requirement is that the model needs to train in the data space since conditioning latent models is not trivial (see W1).
>
> > W9: Guidance requires backpropagating through the network. How costly is this relative to other baselines?
>
> [AW9] **Table J** shows that Harpoon takes roughly **2x** longer than the version without tangent-gradient updates (DiffPuter), due to the gradient computations, which doubles the number of function calls to the network. Yet it remains very fast (**<5 seconds** on largest dataset Adult), and is relatively much faster than GReaT or Miracle. Full runtime comparisons on more datasets, including standard deviations, are provided here [ https://anonymous.4open.science/r/ManifoldTabularImputation-44E4/experiments/runtime.csv ].
>
> |Dataset|Harpoon|DiffPuter|GReaT|Remasker|Miracle|Gain|
> |-|-|-|-|-|-|-|
> |Adult|4.426|2.830|65.472|2.676|33.032|0.138|
> |Bean|2.079|1.317|75.285|0.290|19.859|0.140|
> |California|2.884|1.858|25.050|0.289|21.314|0.147|
> |Default|4.047|2.622|202.711|0.329|44.537|0.148|
> |Gesture|1.543|0.984|131.717|0.297|23.086|0.138|
> |Letter|2.804|1.787|103.679|0.301|26.391|0.138|
> |Magic|2.65 |1.680|19.837|0.296|19.989|0.138|
> |Shoppers|1.890|1.193|51.135|0.280|18.274|0.135|
>
>
> **Table J. Sampling runtime (seconds) for imputation task MAR under 0.25 mask ratio, averaged over 5 trials.**
>
> > W10: It should be highlighted that the approach is more similar to penalization than imposing hard constraints.
>
>
> [AW10] We will clarify this in the paper as suggested. In practice, once the inference-time loss is sufficiently close to zero, we can enforce the constraints in a hard manner without changing the results, since the outputs almost satisfy the constraints anyway. This means that Harpoon can be used as a soft-constraint precursor, which can later be complemented by hard constraints upon convergence. **Table H (W4)** shows that the recovery loss is very small by just using soft constraints.
>
> > W11: The authors do not discuss how their approach differs from classifier-guidance when using similar losses.
>
> [AW11] We will add these details in the paper. Classifier guidance requires a pre-trained classifier to provide gradients for the conditional likelihood of a discrete label. For continuous or partially observed features, it is unclear what the “label” would be. Moreover, each conditioned feature would require a separate classifier, which does not scale to conditions on several features. In contrast, Harpoon flexibly conditions on several continuous and discrete features at inference, broadening applicability.

---

> ### Author Response · Authors · 2025-11-16
> **Response by Authors (Questions Q1-Q6)**
>
> > Q1: Could this manifold guidance lead to a skewed (conditional) data distribution?
>
> [AQ1] Results in **Table G, W3** indicate that KS and alpha scores are generally high. However, in some cases (Shoppers), we see a low alpha precision and high KS, indicating that the marginal distributions match, but the joint distributions may be skewed under situations that stress test by constraining on the tail end of distributions. Full results are here [ https://anonymous.4open.science/r/ManifoldTabularImputation-44E4/experiments/general_constraints_updated_utility.csv ].
>
> > Q2: The guidance mechanism only enforces soft constraints. Could it lead to deviations from the partially observed information?
>
> [AQ2] We refer the reviewer to our response in **W10**. Harpoon can be used as a soft-constraint precursor. Once the loss converges close enough to zero (see **Table H W4**), the constraints are nearly satisfied, so hard constraints can be safely injected at this point.
>
> > Q3: Does the guidance also work for latent diffusion models? Why is skipping shells during denoising a problem? Is this an artifact from a incable?
>
> [AQ3] No, Harpoon assumes a data space model is used (See **W1**). This is because conditioning a latent model requires translating a data-space condition to its latent version which is not trivial. We will state this in the paper.
>
> Regarding the shell skips, our motivation is based off (*Chung, Hyungjin, et al. "Improving diffusion models for inverse problems using manifold constraints." Advances in Neural Information Processing Systems 35 (2022), 25683-25696*). Since the model is trained on noisy samples from shells with matching timesteps during training $(x_t, t)$, misaligning the two at inference $(x_{i\neq t}, t)$ can lead to incorrect estimates.
>
> > Q4: How does your guidance generalize to high dimensional features, since one-hot encoding blows up when applied to 100s or 1000s of categories.
>
> [AQ4] The size of features depends on the encoding scheme. For very large cardinalities, alternatives such as integer or binary encoding can be used, with the tradeoff that unwanted semantic relationships may be introduced between categories. Harpoon can also flexibly work with these schemes, as shown in **Tables 12 and 13 in Appendix H2** (one hot vs integer encoding).
>
>
> > Q5: Can relative or logical constraints be accommodated? e.g., a data table of order and delivery dates.
>
> [AQ5] Yes, such constraints can indeed be handled. Any constraint expressed as a differentiable loss can be incorporated into Harpoon’s guidance framework, and we show results on logical **conjunctions** of range and categorical constraints in **Table 3** of the paper. The loss for constraints such as $A < B$ can be written as $ReLU(A-B)$, similar to the other inequality constraints.
>
> We also empirically verify results for **disjunctive** constraints in **Table K**, by formulating the loss for a disjunction e.g. ($A < a$ OR $B > b$) as a product, i.e., $ReLU(A-a) * ReLU(b-B)$. The key takeaway is that we see a low violation rate for Harpoon even under disjunctions, indicating the generalisability of the method to logical constraints. Full results are here (OR constraints) [ https://anonymous.4open.science/r/ManifoldTabularImputation-44E4/experiments/general_constraints_updated_utility.csv ].
>
> |Dataset|Method|Avg xgb utility|Avg Alpha-P|Avg ViolationAcc|Avg ks|Avg detect|Avg identifiability|
> |-|-|-|-|-|-|-|-|
> ||GReaT| 0.940|0.882|67.140|0.972|0.897|0.242|
> |adult|Harpoon| 0.934|0.870|3.038|0.949|0.921|0.232|
> ||DiffPuter| 0.931|0.876|63.219|0.943|0.925|0.220|
> ||GReaT| 0.838|0.904|24.039|0.930|0.843|0.237|
> |default|Harpoon| 0.813|0.649|4.231|0.924|0.945|0.199|
> ||DiffPuter| 0.819|0.880|23.034|0.944|0.796|0.371|
> ||GReaT| 0.347|0.821|65.756|0.922|0.980|0.141|
> |shoppers|Harpoon| 0.886|0.471|20.200|0.913|0.903|0.369|
> ||DiffPuter| 0.880|0.535|50.560|0.902|0.932|0.304|
>
> **Table K. Results on disjunctions of range and categorical constraints.**
>
>
> > Q6: Figure 2 b) and c) give a good intuition of moving a sample along a shell. Can there be situations in practice where this fails, such as disconnected manifolds?
>
> [AQ6] Yes, the model can also work for constraints that disconnect (see **Table K, Q5**). Disjunctive constraints, like (*colour='red'* OR *colour='blue'*) are one such example that create disconnected submanifolds. In such a cases, moving towards **any** of the regions yields a valid result. However, diversity may be lower (high detectability) since the outputs likely concentrate on regions that are larger or lie closer to $x_t$, due to the stronger pull. See the new illustration here [ https://anonymous.4open.science/r/ManifoldTabularImputation-44E4/new_figures/disconnect.pdf ].

---

> ### Author Response · Authors · 2025-11-16
> **Response by Authors (Q7-Q9)**
>
> > Q7: Are you using the QuantileTransformer to transform the continuous features for the TabDDPM model.
>
> [AQ7] We do not use TabDDPM. We use DiffPuter's codebase, which preprocesses using a StandardScaler. **Table L** below compares the imputation performance of Harpoon (MAR, 0.25 missing ratio) under the standard Z-scaling and quantile scaling ( https://scikit-learn.org/stable/modules/generated/sklearn.preprocessing.QuantileTransformer.html ).
>
> We observe that MSE can increase drastically under quantile scaling (see Default and Shoppers), while categorical accuracies do not change much. This happens because the quantile transformer reshapes continuous features so that their values follow a uniform (or normal) distribution. For datasets with highly skewed, sparse, or irregular continuous features, this transformation can become very nonlinear, making nearby values in the original space map to far-apart values in the transformed space. Since the diffusion model produces outputs in this transformed space, even small prediction errors can blow up after applying the inverse transform back to the original scale. In contrast, Z-scaling preserves the relative geometry of the data, resulting in much more stable performance. Full results are in the revised text (**Table 16, Appendix I.4**).
>
> |Dataset|Scaling|Avg. MSE|Avg. Acc.|
> |-|-|-|-|
> |Adult|Standard|0.99|69.44|
> ||Quantile|1.03|66.60|
> ||
> |Default|Standard|0.51|73.49|
> ||Quantile|12.19|74.88|
> ||
> |Shoppers|Standard|0.73|51.17|
> ||Quantile|5.79|50.86|
>
> **Table L. Imputation MSE and accuracy (discrete feats.) under MAR 0.25 mask.**
>
> > Q8: In line 420: "not just the constrained ones [...]", do you actually mean the unconstrained features?
>
> [AQ8] We thank the reviewer for indicating this. It should indeed be unconstrained, and we will make the correction.
>
> > Q9: The red and green lines in Figure 3 are not visible due to overlaps.
>
> [AQ9] To improve clarity, we have created a separate plot to show the red and green lines without overlaps. We have also normalised the x-axis by dividing the denoising step by the total number of steps, to represent a scale in [0,1]. See here [ https://anonymous.4open.science/r/ManifoldTabularImputation-44E4/experiments/tubular_region_plots/gradient_angles_adult_double.pdf ].

---

> > ### Comment · Reviewer_1jxC · 2025-11-25
> >
> > I thank the authors for their time investment in providing comprehensive answers and updating their document. Below are some additional comments.
> >
> > > We will add more background details on diffusion and tabular diffusion models. Harpoon's guidance is currently limited to data-space models
> >
> > This is what I thought. Please ensure that this point is made clear in the paper. Most diffusion models for tabular data are in data space, so this restriction should not be a problem.
> >
> > I also checked lines 86-89 in the updated document. I am sorry to be a bit pedantic here but I do know many of the recent papers on tabular diffusion models. Therefore, I want to highlight that CDTD does not use a *pre-trained* encoder but only learns an embedding alongside the diffusion model. I am not sure if this constitutes a data-space or latent-space model from your perspective, though. The case for the TabSyn model on the other hand is clear and correct in the text.
> >
> > > **Table G** shows the results for range constraints with additional metrics using Synthcity
> >
> > Thank you for the additional results. From the paper, it does not become clear how you estimate identifiability.
> >
> > > [AW6] We have added a new figure here
> >
> > Thank you very much, this figure is much more clear.
> >
> > > [AW7] **Table I** presents the ablation results using a linearly increasing guidance strength as decreases.
> >
> > Thank you again! I think this was an important to check. For continuous features there may even be cases where a non-fixed guidance would be preferred.
> >
> > > [AW9] **Table J** shows that Harpoon takes roughly **2x** longer than the version without tangent-gradient updates (DiffPuter), due to the gradient computations, which doubles the number of function calls to the network.
> >
> > This should be mentioned in the results section, not just in the appendix. It is important to contextualize the results. I do not think that this is a problem or diminishes your method, so there is no reason to hide it.
> >
> > > [AW10] We will clarify this in the paper as suggested. In practice, once the inference-time loss is sufficiently close to zero, we can enforce the constraints in a hard manner without changing the results, since the outputs almost satisfy the constraints anyway. This means that Harpoon can be used as a soft-constraint precursor, which can later be complemented by hard constraints upon convergence.
> >
> > Thank you for the explanation.  This interpretation makes sense.
> >
> > > [AW11] We will add these details in the paper. Classifier guidance requires a pre-trained classifier
> >
> > I think this provides some additional motivation for why your method is important. I see that you added a section to the introduction regarding classifier-guidance as well. You might want to highlight that you would actually need to train a classifier to make the prediction at all $t$! So this is actually a more difficult task than it seems.
> >
> > > [AQ1] Results in **Table G, W3** indicate that KS and alpha scores are generally high. However, in some cases (Shoppers), we see a low alpha precision and high KS, indicating that the marginal distributions match, but the joint distributions may be skewed under situations that stress test by constraining on the tail end of distributions.
> >
> > Are these results also available in the paper? Is this result expected when using manifold guidance?
> >
> > > [AQ5] Yes, such constraints can indeed be handled. Any constraint expressed as a differentiable loss can be incorporated into Harpoon’s guidance framework,
> >
> > Very interesting! Thank you!
> >
> > > [AQ6] Yes, the model can also work for constraints that disconnect (see **Table K, Q5**). [....] However, diversity may be lower (high detectability) since the outputs likely concentrate on regions that are larger or lie closer to , due to the stronger pull.
> >
> > That makes sense. Please make sure to mention this in the paper to future users are not caught off guard.
> >
> > > Since the diffusion model produces outputs in this transformed space, even small prediction errors can blow up after applying the inverse transform back to the original scale. In contrast, Z-scaling preserves the relative geometry of the data,
> >
> > Makes perfect sense, although these models should be powerful enough such that such high MSEs for default and shoppers (under the Quantile transformation) do not occur. However, I discovered qualitatively similar results in practice. A common solution is also to apply standardization after the Quantile transformation.
> >
> > ----
> >
> > I thank the authors for their extensive efforts in acknowledging and incorporating my feedback. I kindly ask the authors to (a) mention the increased running time in the main part of the paper, preferably in the result section and (b) mention in their introduction that the method works only on diffusion models defined in data-space. In response, I am going to increase my score to an 8 and wish the authors all the best.

---

> ### Author Response · Authors · 2025-11-26
> **Thank you**
>
> We are very greatful for your constructive and detailed feedback! We have made additional revisions to the draft based on your recommendations. Here is a summary:
>
> 1) We have explicitly mentioned the data-space requirement for our model in both the introduction as well as in the background section (before Geometry of image diffusion).
> 2) For space reasons, we have added a smaller table of runtime results (3 datasets) and have mentioned that the full set of results are available in the appendix.
>
>
> Clarification:
> > [AQ1] Results in Table G, W3 indicate that KS and alpha scores are generally high. However, in some cases (Shoppers), we see a low alpha precision and high KS, indicating that the marginal distributions match, but the joint distributions may be skewed under situations that stress test by constraining on the tail end of distributions.
> Are these results also available in the paper? Is this result expected when using manifold guidance?
>
> Yes, we have added these results using additional metrics to Table 18 in Appendix I.5. We believe this skewness occurs because the model ends up prioritising some constraint regions over others, similar to the disjunction scenario with disconnections. Especially when the representative training set is already small, because this makes the model more likely to "greedily" pick the closest solution to meet the constraints.
>
> We would like to thank you once again for the valuable feedback and for increasing your score to 8!

---

### Official Review · Reviewer_c4MN · 2025-10-30

**Soundness:** 3
**Presentation:** 4
**Contribution:** 3
**Rating:** 4
**Confidence:** 4

**Summary:**

This paper introduces a method that can perform conditional tabular diffusion named HARPOON. It can generate new tabular data with specific conditions. In the theory contribution section, the authors show: 1. diffusion de-noiser acts as orthogonal projectors to data manifold 2. any differentiable inference-time loss has gradient in the tangent space of the manifold and therefore updates using this gradient can preserve realism. HARPOON is a method of combining unconditional de-noising step and the constraint-guided tangent gradient step. The experiments show that HARPOON outperforms other baseline methods.

**Strengths:**

1. The paper is well written and provides both theoretical and experimental contributions
2. The theory is novel and is the first to use diffusion model's orthogonal projection to manifolds in tabular setting
3. HARPOON can handle mixed data types and has much lower constraint violation rate when doing conditional generation

**Weaknesses:**

1. Computation time might be an issue but this is not discussed in the experiments
2. The utility, fidelity and privacy aspects of the generated tabular data is not discussed. It would be great to see where HARPOON stands among these 3 aspects.

**Questions:**

1. Does the tabular data generated from HARPOON improve the downstream model?
2. What does the run time of HARPOON look like against other methods?

---

> ### Author Response · Authors · 2025-11-16
> **Response by Authors**
>
> ### Weaknesses
> > W1: Computation time might be an issue but this is not discussed in the experiments
>
>
>
> [AW1] **Table D** shows that Harpoon takes roughly **2x** longer than the version without tangent-gradient updates (DiffPuter), due to the gradient computations, which doubles the number of function calls to the model. Yet it remains very fast (**<5** seconds at worst), and is much faster than GReaT or Miracle. Full runtime comparisons on more datasets, including standard deviations, are provided here [ https://anonymous.4open.science/r/ManifoldTabularImputation-44E4/experiments/runtime.csv ].
>
> | Dataset  | Harpoon | DiffPuter | GReaT | Remasker | Miracle | Gain |
> | --- |---|---|---|---|---|---|
> | Adult | 4.426 | 2.830 | 65.472 | 2.676 | 33.032 | 0.138 |
> | Bean | 2.079 | 1.317 | 75.285 | 0.290 | 19.859 | 0.140 |
> | California | 2.884 | 1.858 | 25.050 | 0.289 | 21.314 | 0.147 |
> | Default | 4.047 | 2.622 | 202.711 | 0.329 | 44.537 | 0.148 |
> | Gesture | 1.543 | 0.984 | 131.717 | 0.297 | 23.086 | 0.138 |
> | Letter | 2.804 | 1.787 | 103.679 | 0.301 | 26.391 | 0.138 |
> | Magic | 2.652 | 1.680 | 19.837 | 0.296 | 19.989 | 0.138 |
> | Shoppers | 1.890 | 1.193 | 51.135 | 0.280 | 18.274 | 0.135 |
>
>
> **Table D. Sampling runtime (seconds) for imputation task MAR under 0.25 mask ratio, averaged over 5 trials.**
>
> > W2: The utility, fidelity and privacy aspects of the generated tabular data is not discussed.
>
> [AW2] **Table E** shows the results for range constraints with additional metrics from Synthcity [ https://github.com/vanderschaarlab/synthcity ] . Full results are here [ https://anonymous.4open.science/r/ManifoldTabularImputation-44E4/experiments/general_constraints_updated_utility.csv ]. High KS scores indicate that Harpoon captures the marginal distributions of real data well. Low identifiability indicates that the model does not memorise real samples, yielding better privacy. While high detectability indicates that the models have selective bias, we still see a high downstream utility of Harpoon  (XGBoost classifier).
>
> |Dataset|Method|Avg xgb utility|Avg Alpha-P|Avg ViolationAcc|Avg ks|Avg detect|Avg identifiability|
> |-|-|-|-|-|-|-|-|
> ||GReaT| 0.947|0.922|83.577|0.962|0.953|0.196|
> |adult|Harpoon| 0.945|0.918|3.065|0.951|0.905|0.256|
> ||DiffPuter| 0.941|0.829|73.861|0.937|0.944|0.169|
> ||GReaT| 0.696|0.670|95.366|0.915|0.987|0.077|
> |default|Harpoon| 0.855|0.787|7.782|0.938|0.886|0.307|
> ||DiffPuter| 0.778|0.649|88.525|0.935|0.942|0.155|
> ||GReaT| 0.348|0.877|96.525|0.911|0.994|0.071|
> |shoppers|Harpoon| 0.886|0.604|20.545|0.922|0.874|0.436|
> ||DiffPuter| 0.892|0.684|76.687|0.907|0.937|0.292|
>
> **Table E. Additional metrics on range constraints averaged over 5 trials.**
>
> ### Questions
> > Q1: Does the tabular data generated from HARPOON improve the downstream model?
>
> [AQ1] Yes, Harpoon shows strong accuracy (85%+) using a downstream XGBoost classifier (see W2).
>
> > Q2: What does the run time of HARPOON look like against other methods?
>
> [AQ2] Refer to our response to W1.

---

### Official Review · Reviewer_kUVE · 2025-11-01

**Soundness:** 3
**Presentation:** 3
**Contribution:** 3
**Rating:** 8
**Confidence:** 3

**Summary:**

The paper proposes a new way to guide diffusion models for tabular generation at inference time. In particular, they show how at every time step $t$ most of the datapoints lie on a shell of dimension $d-1$ ($d$ being the original dimensionality of the datapoints) and that while the denoising process creates an orthogonal projection onto the shell of the datapoint $x_t$, the gradient of any loss function defined  starting from a condition $c$ is tangent to such shell. Exploiting these geometric results, they are able to create a new procedure for guiding the diffusion process at inference time, where essentially they interleave the diffusion process which (as they nicely put it) acts as a "compass" towards the manifold of the original datapoints and the update wrt the loss function that encodes the constraints which moves along the shell to guide the model towards the right region of the manifold.

**Strengths:**

The paper is very well written and provides a lot of intuitions about the results they propose. I really appreciated the visualisations of the gradients and the updates.

The experimental analysis is extensive and with very positive results.

The authors give a very nice geometric explanation of why their method works.

**Note:** It is difficult for me to assess the novelty of this work wrt the previous works on diffusion models as I am not familiar with them.

**Weaknesses:**

1.  The authors do not report the sampling generation time. As this is an important metric for tabular data generation, it would be nice to have it

2. In theorem 3.2 $\mathcal{C}$ is not defined. Also it is not clear which conditions we have on $\mathcal{C}$. Can it really be any arbitrary information? For example (Stoian & Giunchiglia, 2025) has extended the work cited in your paper to constraints expressed as disjunctions over linear inequalities. This defines non-convex and disconnected spaces hence violating your assumption 1. Would this time of conditioning be allowed? What about polynomials?

3. A better definition of the alpha metric is needed.



Minor things:

1. Citation for Borisov et al. is missing the year
2. Sometimes equation is written with capitol E and sometimes with lower case e

References:

Stoian & Giunchiglia. Beyond the convexity assumption: Realistic tabular data generation under quantifier-free real linear constraints, ICLR, 2025.

**Questions:**

1. In order to impose the linear inequality constraints you had to devise an ad-hoc loss function. Do you see this as a possible bottleneck for the widespread application of your solution?

2. Aside from categorical constraints, it is very reasonable to assume that tabular data have disconnected support. This makes me wonder how realistic is assumption 1 and also how important it is that assumption 1 is met in practice. Would it be possible to have an ablation study where a multiple datasets are created with each of them violating the assumption in different degrees and then studying how the method performs on them?

---

> ### Author Response · Authors · 2025-11-16
> **Response by Authors (Weaknesses)**
>
> ### Weaknesses
> > W1: The authors do not report the sampling generation time.
>
> [AW1] **Table A** below has runtime costs. We use a 12-core AMD Ryzen 9 5900 processor with an NVIDIA RTX 3090 GPU, CUDA 12.1, Ubuntu 20.04.6, and Pytorch 2.2.1 with Python 3.12.
>
> Results show that Harpoon takes roughly **2x** longer than its "unconditional" proxy (DiffPuter). This is expected since the backward pass for guidance doubles the number of neural network calls. Still, the total runtime remains **below 5 seconds**, substantially faster than Miracle and GReaT. The full results are here [ https://anonymous.4open.science/r/ManifoldTabularImputation-44E4/experiments/runtime.csv ].
>
> |Dataset|Harpoon|DiffPuter|GReaT|Remasker|Miracle| Gain|
> |-|-|-|-|-|-|-|
> |Adult|4.426|2.830|65.472|2.676|33.032|0.138|
> |Bean|2.079|1.317|75.285|0.290|19.859|0.140|
> |California|2.884|1.858|25.050|0.289|21.314|0.147|
> |Default|4.047|2.622|202.711|0.329|44.537|0.148|
> |Gesture|1.543|0.984|131.717 |0.297|23.086|0.138|
> |Letter|2.804|1.787|103.679|0.301|26.391|0.138|
> |Magic|2.652|1.680|19.837|0.296|19.989|0.138|
> |Shoppers|1.890|1.193|51.135|0.280|18.274|0.135|
>
>
> **Table A. Sampling runtime (seconds) for MAR imputation under 0.25 ratio, averaged over 5 trials.**
>
>
> > W2: In theorem 3.2 $C$ is not defined. Can it work for disconnected manifolds, e.g. disjunctive constraints?
>
> [AW2] $C$ is assumed to encode constraints that can be represented using a differentiable loss. We theoretically prove that the gradient lies in the tangent space as long as the loss is differentiable. Disjunctive constraints result in disconnected submanifolds within the larger surface. In practice, $x_t$ would converge to the region that lies closer in geometric space (see [ https://anonymous.4open.science/r/ManifoldTabularImputation-44E4/new_figures/disconnect.pdf ]). For example the constraint ($A < a$ OR $B > b$) can be captured by multiplying the individual constraint losses as follows.
>
> $loss = ReLU(A - a) * ReLU(b - B)$.
>
> This loss drops to zero when either constraint is satisfied. **Table B** below empirically evaluates Harpoon on disjunctions of categorical and range constraints. We see that Harpoon achieves the **lowest violation rate**, indicating successful guidance of samples towards valid regions. High KS scores indicate that marginal distributions are preserved. Although alpha precision decreases sometimes, it is expected since the model may converge to some modes more readily than others when there are multiple feasible regions. Low identifiability shows the model is not memorising training examples, while higher detection scores indicate some bias introduced by the disconnect. Still, high utility scores (XGBoost classifier), indicate that downstream performance is not affected. Full results are here (OR constraints)[ https://anonymous.4open.science/r/ManifoldTabularImputation-44E4/experiments/general_constraints_updated_utility.csv ].
>
> |Dataset|Method|Avg xgb utility|Avg Alpha-P|Avg ViolationAcc|Avg ks|Avg detect|Avg identifiability|
> |-|-|-|-|-|-|-|-|
> ||GReaT| 0.940|0.882|67.140|0.972|0.897|0.242|
> |adult|Harpoon| 0.934|0.870|3.038|0.949|0.921|0.232|
> ||DiffPuter| 0.931|0.876|63.219|0.943|0.925|0.220|
> ||GReaT| 0.838|0.904|24.039|0.930|0.843|0.237|
> |default|Harpoon| 0.813|0.649|4.231|0.924|0.945|0.199|
> ||DiffPuter| 0.819|0.880|23.034|0.944|0.796|0.371|
> ||GReaT| 0.347|0.821|65.756|0.922|0.980|0.141|
> |shoppers|Harpoon| 0.886|0.471|20.200|0.913|0.903|0.369|
> ||DiffPuter| 0.880|0.535|50.560|0.902|0.932|0.304|
>
> **Table B. Results on disjunctions of range and categorical constraints, averaged over 5 trials.**
>
>
> > W3: A better definition of the alpha metric is needed.
>
> [AW3] We will add a clearer definition of the metric in the paper. Intuitively, it captures whether points fall in the same geometric regions as real data. It is the fraction of synthetic samples that lie close to the most typical fraction $\alpha$ of real samples (*Alaa, Ahmed, et al. "How faithful is your synthetic data? sample-level metrics for evaluating and auditing generative models." International conference on machine learning. PMLR, 2022*).

---

> ### Author Response · Authors · 2025-11-16
> **Response by Authors (Minor comments and Questions)**
>
> ### Minor comments
> > M1: Citation for Borisov et al. is missing the year
>
> [AM1] We thank the reviewer for pointing this out and we will add the year.
>
> > M2: Sometimes equation is written with capitol E and sometimes with lower case e.
>
> [AM2] Thanks for pointing it out. We will make it consistent throughout the text.
>
> ### Questions
> > Q1: The loss function for imposing linear inequality constraints is ad-hoc. Is this as a possible bottleneck for the widespread application of your solution?
>
> [AQ1] We theoretically prove that we can guide the outputs towards any constraint that can be represented using a differentiable loss. Our objective is not to exhaustively catalogue loss functions for all types of constraints. Harpoon is rather a guidance tool for an end user to specify a differentiable loss for their specific task. As a proof-of-concept, we evaluated on imputation and linear inequality constraints by reusing existing differentiable losses from prior work (*Donti, Priya L., David Rolnick, and J. Zico Kolter. "DC3: A learning method for optimization with hard constraints." arXiv preprint arXiv:2104.12225 (2021)*).
>
> > Q2: How realistic is assumption 1 and hold in practice? Possible to study the effect of varying degrees of disconnect on performance?
>
> [AQ2] The disconnectedness of the support depends on the encoding scheme. If disconnection is suspected, integer encodings for discrete features followed by standardisation can provide a continuous approximation that locally satisfies Assumption 1, serving as a fallback option. Even if Assumption 1 is not strictly met, we conjecture that diffusion models implicitly learn a smooth approximation of the manifold through their stochastic, sampling-based training.
>
> **Table C** below compares Harpoon under different degrees of disconnectedness: conditions in isolation, conjunctions, and disjunctions. Most metrics generally stay stable across settings. But disjunctions can sometimes lower the alpha fidelity, since the model may be skewed to favour one condition over the other (see W2). Notably, the violation rate from disjunctions is consistently lower than the worst  condition in isolation, possibly because the feasible region increases. Full results are here [ https://anonymous.4open.science/r/ManifoldTabularImputation-44E4/experiments/general_constraints_updated_utility.csv ].
>
> |Dataset|Constraint|Avg xgb utility|Avg Alpha-P|Avg ViolationAcc|Avg ks|Avg detect|Avg identifiability|
> |-|-|-|-|-|-|-|-|
> |adult|Range| 0.945|0.918|3.065|0.951|0.905|0.256|
> ||Category| 0.880|0.857|2.178|0.967|0.733|0.459|
> ||Range AND Category| 0.908|0.820|3.371|0.963|0.642|0.470|
> ||Range OR Category| 0.934|0.870|3.038|0.949|0.921|0.232|
> |default|Range| 0.855|0.787|7.782|0.938|0.886|0.307|
> ||Category| 0.833|0.920|0.906|0.951|0.809|0.396|
> ||Range AND Category| 0.809|0.872|2.984|0.945|0.816|0.375|
> ||Range OR Category| 0.813|0.649|4.231|0.924|0.945|0.199|
> |shoppers|Range| 0.886|0.604|20.545|0.922|0.874|0.436|
> ||Category| 0.877|0.708|0.000|0.899|0.865|0.469|
> ||Range AND Category| 0.853|0.809|12.053|0.895|0.816|0.530|
> ||Range OR Category| 0.886|0.471|20.200|0.913|0.903|0.369|
>
> **Table C. Harpoon guidance under varying disconnectedness.**

---

> > ### Comment · Reviewer_kUVE · 2025-11-27
> >
> > I was already very happy with the paper so I will simply maintain my rating.

---

> > > ### Author Response · Authors · 2025-11-28
> > > **Thank you**
> > >
> > > Dear reviewer kUVE,
> > >
> > > We are grateful to you for retaining your score of 8 and your constructive feedback which helped us increase the quality of our paper significantly.
> > >
> > > Best regards,
> > >
> > > Authors of Harpoon

---

### Author Response · Authors · 2025-11-16
**Response Summary by Authors**

## Main comment

We thank all reviewers for the valuable feedback. We conducted additional experiments and added new figures to strengthen the paper. All the updates will be added to the revised paper.

*(NOTE: Multiple reviewers sometimes had overlapping concerns. We have indicated these below and have copy-pasted the result tables for each reviewer individually for ease of read*).

**Summary of additional experiments:**
* **Sampling runtimes** (Reviewers kUVE, c4MN , 1jxC). **Tables A, D, J**.
* **Results on disjunctions of range and categorical constraints** (Reviewers kUVE, 1jxC). To test guidance under disconnected submanifolds. **Tables B, K**.
* **Effect of varying disconnectedness of constraints** (Reviewer kUVE). **Table C**.
* **Results with additional metrics on utility, fidelity, and privacy** (Reviewers  c4MN, 1jxC). **Tables E, G**
* **Results with lower missing ratios** (Reviewer 1jxC). **Table F**.
* **Recovery under large number of constraints** (Reviewer 1jxC). **Table H**.
* **Results on linear guidance schedule** (Reviewer 1jxC).**Table I**.
* **Results with quantile transformer** (Reviewer 1jxC).**Table L**.

**Additional figures**

* **Disconnected (sub)manifolds** (Reviewers kUVE, 1jxC).
* **Improved spotlight figure** (Reviewer 1jxC).
* **Updated empirical orthogonality plot** (Reviewer 1jxC).

We hope these additions address the concerns raised by the reviewers.

---

> ### Author Response · Authors · 2025-11-24
> **Revised draft**
>
> We have now updated the draft and rebuttal text with the new experimental results, figures, and additional textual descriptions (indicated in blue). Please let us know if you require any further clarifications.

---

### Author Response · Authors · 2025-11-28
**Comment from the Authors to whoever will be the new area chair**

Dear (Newly) Assigned AC,

We have been informed that ICLR is going to be reverting the scores during the discussion phase. We would like to summarise to you about the discussions we had with reviewers + the score/confidence changes made.

1) **Reviewer kUVE**: Original rating = (**8, confidence 3**). During discussion phase: **Retained the score** (**8, confidence 3**)
Official Comment by Reviewer kUVE 27 Nov 2025
: "*I was already very happy with the paper so I will simply maintain my rating.*"

2) **Reviewer c4NM**: Original rating = (**4, confidence 4**). **Did not respond to our rebuttal during the discussion phase.**

3) **Reviewer 1jxC**: Original rating = (**4, confidence 3**). During discussion phase: **increased the score and confidence significantly** (**8, confidence 4**)
Official Comment by Reviewer 1jxC 25 Nov 2025
: "*I thank the authors for their extensive efforts in acknowledging and incorporating my feedback. I kindly ask the authors to (a) mention the increased running time in the main part of the paper, preferably in the result section and (b) mention in their introduction that the method works only on diffusion models defined in data-space. In response, I am going to increase my score to an 8 and wish the authors all the best.*"

We hope this update aids you in making an informed decision regarding our paper.

Best regards,

Authors of Harpoon

---

### Meta-Review · Area_Chair_kUeN · 2026-01-05

**Summary:**

This paper introduces Harpoon, a training-free guided generation method for tabular data.

I understand that [1] and [2] rely on linear-manifold assumptions, and Harpoon extends the theory beyond that setting. That said, the core sampling procedure follows a standard pattern: an unconditional denoise step interleaved with a gradient step on an inference-time loss. The submission does not empirically compare against established training-free, loss-guided diffusion baselines in the spirit of [1] and [2] using the same constraint losses, which makes it difficult to attribute the gains specifically to the proposed manifold or tangential guidance rather than to generic gradient-based guidance.

I strongly suggest the authors (i) explicitly clarify the algorithmic differences and the conceptual distinctions between Harpoon and prior guidance approaches such as [1,2], and (ii) if feasible, include a numerical comparison to these or closely matched training-free gradient-guidance baselines under the same constraint losses and evaluation protocol.

Although, to me, the methodological novelty appears limited, the paper is well-written and reports strong empirical results, and most concerns raised by the reviewers appear to be addressed. Therefore, I respect the overall reviewer feedback and recommend a **weak accept**.



----
[1] Chung, Hyungjin, et al. "Improving diffusion models for inverse problems using manifold constraints." Advances in Neural Information Processing Systems 35 (2022): 25683-25696.

[2] He, Yutong, et al. "Manifold preserving guided diffusion." arXiv preprint arXiv:2311.16424 (2023).

**Reviewer Concerns:**

- Reviewer kUVE’s main concern is the generation-time computational cost.

- Reviewer c4MN raises two main concerns: (i) generation-time computational cost may be an issue but is not evaluated or discussed in the experiments, and (ii) the utility, fidelity, and privacy of the generated tabular data are not analyzed.

- Reviewer 1jxC raises mainly five concerns: (i) the missingness rates of 0.5 and 0.75 are very high, and it is unclear how the method performs under more moderate missingness; (ii) it would be useful to include metrics that assess the joint distribution more holistically; (iii) the reviewer asks how performance changes as the number of constraints increases, and whether the method can recover a single valid observation in the extreme case; (iv) beyond one-hot encoding, the paper does not include tabular-data-specific modeling; and (v) the reviewer asks whether the orthogonality assumption for all timesteps can be relaxed by using time-dependent guidance strength, for example increasing guidance as (t) approaches 0.

**Reviewer Scores:**

I believe the authors have addressed most of these concerns by adding additional experiments and clarifications. Therefore, I believe the scores should either remain unchanged or be increased. In particular, Reviewer 1jxC stated that they would update their score from 4 (confidence: 3) to 8 (confidence: 4).

---

### Decision · Program_Chairs · 2026-01-26

Accept (Poster)